# Metropolis Adjusted Microcanonical Hamiltonian Monte Carlo

**Jakob Robnik**
Physics Department,
University of California at Berkeley,
Berkeley, CA 94720, USA
jakob_robnik@berkeley.edu

**Reuben Cohn-Gordon**
Physics Department,
University of California at Berkeley,
Berkeley, CA 94720, USA
reubenharry@gmail.com

**Uroš Seljak**
Physics Department,
University of California at Berkeley
and Lawrence Berkeley National Laboratory, Berkeley,
Berkeley, CA 94720, USA
useljak@berkeley.edu

## Abstract

Sampling from high dimensional distributions is a computational bottleneck in many scientific applications. Hamiltonian Monte Carlo (HMC), and in particular the No-U-Turn Sampler (NUTS), are widely used, yet they struggle on problems with a very large number of parameters or a complicated geometry. Microcanonical Langevin Monte Carlo (MCLMC) has been recently proposed as an alternative which shows striking gains in efficiency over NUTS, especially for high-dimensional problems. However, it produces biased samples, with a bias that is hard to control in general. We introduce the *Metropolis-Adjusted Microcanonical sampler* (MAMS), which relies on the same dynamics as MCLMC, but introduces a Metropolis-Hastings step and thus produces asymptotically unbiased samples. We develop an automated tuning scheme for the hyperparameters of the algorithm, making it applicable out of the box. We demonstrate that MAMS outperforms NUTS across the board on benchmark problems of varying complexity and dimensionality, achieving up to a factor of seven speedup.

## 1 Introduction

Drawing samples from a given probability density $p(\boldsymbol{x})$, for $\boldsymbol{x} \in \mathbb{R}^d$, has applications in a wide range of scientific disciplines, from Bayesian inference for statistics (Štrumbelj et al., 2024; Carpenter et al., 2017) to biology (Gelman and Rubin, 1996), statistical physics (Janke, 2008), quantum mechanics (Gattringer and Lang, 2010) and cosmology (Campagne et al., 2023). From the perspective of a practitioner in these fields, what is often desirable is a *black-box algorithm*, in the sense of taking as input an unnormalized density and returning samples from the corresponding distribution. An important special case is where the density is differentiable (either analytically or by automatic differentiation (Griewank and Walther, 2008)).

Markov Chain Monte Carlo (Metropolis et al., 1953; Hastings, 1970) is a broad class of methods suited to this task, which construct a Markov Chain $\{\boldsymbol{x}_i\}_{i=1}^n$ such that $\boldsymbol{x}_j$ are samples from the target distribution $p$. When the density's gradient is available, Hamiltonian Monte Carlo (HMC) (Duane et al., 1987; Neal, 2011) is a leading method. In particular, the No-U-Turn sampler is a black-box version of HMC, where users do not need to manually select the hyperparameters. It has

39th Conference on Neural Information Processing Systems (NeurIPS 2025).

been implemented in libraries like Stan (Carpenter et al., 2017) and Pyro (Bingham et al., 2019), serving a wide community of scientists.

NUTS is a powerful method, but there have been many attempts in the past decade to replace it with algorithms that can achive the same accuracy at a lower computational cost. One such method is Microcanonical Langevin Monte Carlo (MCLMC) (Robnik et al., 2024); it replaces the HMC dynamics with a velocity-norm preserving dynamics, resulting in a method that is more stable to large gradients. Benchmarking in cosmology (Simon-Onfroy et al., 2025), Bayesian inference (Robnik et al., 2024; Sommer et al., 2024), and field theories (Robnik and Seljak, 2024) suggests MCLMC is a promising candidate to replace NUTS as a go-to gradient-based sampler.

The drawback of MCLMC is that it is biased; that is, the samples it produces do not correspond to samples from the true target distribution $p$, but rather to samples from a nearby distribution $\tilde{p}$. Although this bias is controllable, in many fields, asymptotically unbiased samplers are wanted, limiting the widespread utility of MCLMC. For HMC, this problem is resolved by the use of a Metropolis-Hastings (MH) step (Metropolis et al., 1953; Hastings, 1970), which accepts or rejects proposed moves $\boldsymbol{x} \to \boldsymbol{x}'$ of the Markov chain according to an acceptance $\min\left(1, e^{-W(\boldsymbol{x}', \boldsymbol{x})}\right)$, where

$$e^{-W(\boldsymbol{x}', \boldsymbol{x})} = \frac{p(\boldsymbol{x}')}{p(\boldsymbol{x})} \frac{q(\boldsymbol{x}|\boldsymbol{x}')}{q(\boldsymbol{x}'|\boldsymbol{x})}. \tag{1}$$

But for MCLMC, the corresponding quantity $W$ has not been previously derived. Moreover, an adaptation scheme for choosing the hyperparameters of the algorithm in the MH adjusted case is needed to make the algorithm usable out of the box, so that it can serve the same use cases as NUTS.

**Contributions** In this paper, we derive the acceptance probabilities for microcanonical dynamics with and without Langevin noise in Section 5 and Section 4, respectively. Notably, $W$ turns out to be the energy error induced by discretization of the dynamics, as in HMC. We term the resulting sampler the *Metropolis-Adjusted Microcanonical Sampler* (MAMS), and develop an automatic adaptation scheme (Section 6) to make MAMS applicable without having to specify hyperparameters manually. We test MAMS on standard benchmarks in Section 7 and find that *it outperforms the state-of-the-art HMC with NUTS tuning by a factor of two at worst, and seven at best*. The algorithm is implemented in blackjax (Cabezas et al., 2024), applicable out-of-the-box, and is publicly available, together with documentation and tutorials[1]. The code for reproducing numerical experiments is also available[2].

## 2 Related work

A wide variety of gradient-based samplers have been proposed, including Metropolis Adjusted Langevin trajectories (Riou-Durand and Vogrinc, 2022), generalized HMC (Horowitz, 1991a; Neal, 2020), the Metropolis Adjusted Langevin Algorithm (MALA; Grenander and Miller (1994)), Deterministic Langevin Monte Carlo (Grumitt et al., 2022), Nose-Hoover (Evans and Holian, 1985; Leimkuhler and Reich, 2009), Riemannian HMC (Girolami and Calderhead, 2011), Magnetic HMC (Tripuraneni et al., 2017) and the Barker proposal (Livingstone and Zanella, 2022). Some of these methods come with automatic tuning schemes that make them black-box, for example MALT (Riou-Durand and Vogrinc, 2022; Riou-Durand et al., 2023), generalized HMC (Hoffman et al., 2021) and HMC (Sountsov and Hoffman, 2022; Hoffman et al., 2021), but these schemes are designed for the many-short-chains MCMC regime (Sountsov et al., 2024; Margossian et al., 2024), which we do not consider here [3] To our knowledge, NUTS remains the state-of-the-art black-box method for selecting the trajectory length in HMC-like algorithms for the single chain regime.

The dynamics described by Equation (3) have been independently proposed several times. In computational chemistry, they were derived by constraining Hamiltonian dynamics to have a fixed velocity norm (Tuckerman et al., 2001; Minary et al., 2003) and termed *isokinetic dynamics*. More recently, Steeg and Galstyan (2021) proposed them as a time-rescaling of Hamiltonian dynamics

---

[1]https://blackjax-devs.github.io/sampling-book/algorithms/mclmc.html

[2]https://github.com/reubenharry/sampler-benchmarks

[3]Many-short-chains approach is to run multiple short chains in parallel instead of a single long chain. This regime is interesting when parallel resources are available. However, it is often not applicable, either because one only has a single CPU, or because the parallel resources are needed elsewhere, for example, for parallelizing the model (Gattringer and Lang, 2010) or for performing multiple sampling tasks (Robnik et al., 2024).

with non-standard kinetic energy and no momentum resampling. Robnik et al. (2024) observed that while HMC aims to reach a stationary distribution known in statistical mechanics as the canonical distribution, it is also possible to target what is known as the *microcanonical* distribution, i.e. the delta function at some level set of the energy. The Hamiltonian $H$ must then be chosen carefully to ensure that the position marginal of the microcanonical distribution is the desired target $p$, and one such choice is the Hamiltonian from Steeg and Galstyan (2021). They propose adding velocity resampling every $n$ steps or Langevin noise every step as a method to obtain ergodicity.

In all of these instances, MCLMC has been proposed without Metropolis-Hastings, and as such, has been proposed as a biased sampler. This is the shortcoming that the present work resolves. We refer to our sampler as using *microcanonical* dynamics in reference to previous work on MCLMC, In contrast, we will refer to the dynamics in standard HMC as *canonical* dynamics.

## 3  Technical Preliminaries

**Hamiltonian Monte Carlo**   Let $\mathcal{L}$ be the negative log likelihood of $p$ up to a constant, i.e. $p(\boldsymbol{x}) = e^{-\mathcal{L}(\boldsymbol{x})}/Z$, where $Z = \int e^{-\mathcal{L}(\boldsymbol{x})} d\boldsymbol{x}$. If gradients $\nabla \mathcal{L}(\boldsymbol{x})$ are available and are sufficiently smooth, HMC is the gold standard proposal distribution. In HMC, each parameter $x_i$ has an associated velocity $u_i$. Parameters and their velocities evolve by a set of differential equations

$$\dot{\boldsymbol{x}} = \boldsymbol{u} \qquad \dot{\boldsymbol{u}} = \nabla \log p(\boldsymbol{x}), \tag{2}$$

which are designed to have $p_{HMC}(\boldsymbol{x}, \boldsymbol{u}) = p(\boldsymbol{x})\mathcal{N}(\boldsymbol{u})$ as their stationary distribution. Here $\mathcal{N}$ is the standard normal distribution. Note that the marginal distribution $\int p_{HMC}(\boldsymbol{x}, \boldsymbol{u}) d\boldsymbol{u}$ is equal to $p(\boldsymbol{x})$, the distribution we want to sample from. Thus, sampling from $p(\boldsymbol{x})$ reduces to solving Equation (2). In practice, the dynamics has to be simulated numerically, by iteratively solving for $\boldsymbol{x}$ at fixed $\boldsymbol{u}$ and vice versa, and updating the variables by a time step $\epsilon$ at each iteration. The discrepancy between this approximation and the true dynamics causes the stationary distribution to differ from the target distribution, but this can be corrected by MH; that is, we can use discretized Hamiltonian dynamics as a proposal $q$. Furthermore, to attain ergodicity, the velocities $\boldsymbol{u}$ must be resampled after every $n$ steps.

The resulting algorithm has two hyperparameters: the discretization step size of the dynamics $\epsilon$ and the trajectory length between each resampling $L = n\epsilon$. Choosing good values for these two hyperparameters is crucial (Beskos et al., 2013; Neal, 2011; Betancourt, 2018), and so a practical sampler must also provide a robust *adaptation scheme* for choosing them.

**Microcanonical dynamics**   An alternative to HMC is microcanonical dynamics (Robnik et al., 2024; Tuckerman et al., 2001; Minary et al., 2003; Steeg and Galstyan, 2021) defined by:

$$\dot{\boldsymbol{x}} = \boldsymbol{u} \qquad \dot{\boldsymbol{u}} = (I - \boldsymbol{u}\boldsymbol{u}^T)\nabla \log p(\boldsymbol{x})/(d-1), \tag{3}$$

where $\boldsymbol{u}$ has unit norm which is preserved by the dynamics. The proposed benefit is that the normalization of the velocity makes the dynamics more stable to large gradients. When integrated exactly, these dynamics have $p_{MCLMC}(\boldsymbol{x}, \boldsymbol{u}) = p(\boldsymbol{x})\mathcal{U}_{S^{d-1}}(\boldsymbol{u})$ as a stationary distribution (see Appendix B.4), so that the marginal is still $p(\boldsymbol{x})$. Here $\mathcal{U}_{S^{d-1}}$ is the uniform distribution on the $d-1$ sphere. Robnik et al. (2024) propose using these dynamics without MH in order to *approximately* sample from $p(\boldsymbol{x})$. In this case, the velocity is partially resampled after every step, and the step size of the discretized dynamics is chosen small enough to limit deviation from the target distribution to acceptable levels.

While this algorithm works well in practice when the step size is properly tuned, the numerical integration error is not corrected, resulting in an asymptotic bias which is hard to control. In HMC this is solved by the MH step, which requires calculating $W$, as defined in Equation (1). In this case, $W$ can be easily derived since the integrator is symplectic (volume preserving) and $q(\boldsymbol{x}|\boldsymbol{x}')/q(\boldsymbol{x}'|\boldsymbol{x}) = 1$. The integrator used for microcanonical dynamics is not symplectic, so it not immediately clear how to calculate $W$.

## 4  Metropolis adjustment for canonical and microcanonical dynamics

Both canonical and microcanonical dynamics can be numerically solved by separately solving the differential equation for the parameters $\boldsymbol{x}$, at fixed velocities $\boldsymbol{u}$ and vice versa. For a time interval $\epsilon$,

we refer to the position update as $A_\epsilon(\boldsymbol{x}, \boldsymbol{u})$ and the velocity update as $B_\epsilon(\boldsymbol{x}, \boldsymbol{u})$. The solution of the combined dynamics at time $t = n\epsilon$ is then constructed by a composition of these updates:

$$\varphi = \mathcal{T} \circ \underbrace{\Phi_{t/n} \circ \Phi_{t/n} \circ \cdots \Phi_{t/n}}_{n}, \tag{4}$$

$$\Phi_\epsilon = B_{\epsilon/2} \circ A_\epsilon \circ B_{\epsilon/2}. \tag{5}$$

This is known as the leapfrog (or velocity Verlet) scheme. A final time reversal map $\mathcal{T}(\boldsymbol{x}, \boldsymbol{u}) = (\boldsymbol{x}, -\boldsymbol{u})$, is inserted to ensure the map is an involution, i.e. $\varphi \circ \varphi = id$, where $id$ is the identity map. This is useful in the Metropolis step but does not affect the dynamics in any way, because a full velocity refreshment is performed after the Metropolis step, erasing the effect of time reversal.

Both HMC and MCHMC possess a quantity, which we refer to as energy, which is conserved for exact dynamics, but only approximately conserved by the discrete update from Equation (5). The energy $H$ is composed of two parts, a potential energy $V$ and kinetic energy[4] $K$. The position updates $(\boldsymbol{x}', \boldsymbol{u}) = A_\epsilon(\boldsymbol{x}, \boldsymbol{u})$ change the potential energy by

$$V(A_\epsilon(\boldsymbol{z})) - V(\boldsymbol{z}) = -\log \frac{p(\boldsymbol{x}')}{p(\boldsymbol{x})}, \tag{6}$$

while the velocity updates $(\boldsymbol{x}, \boldsymbol{u}') = B_\epsilon(\boldsymbol{x}, \boldsymbol{u})$ change the kinetic energy by

$$K(B_\epsilon(\boldsymbol{z})) - K(\boldsymbol{z}) = \frac{1}{2} \|\boldsymbol{u}'\|^2 - \frac{1}{2} \|\boldsymbol{u}\|^2 \tag{7}$$

for HMC and by

$$K(B_\epsilon(\boldsymbol{z})) - K(\boldsymbol{z}) = (d-1)\log\{\cosh\delta + \boldsymbol{e} \cdot \boldsymbol{u} \sinh\delta\} \tag{8}$$

for MAMS. Here $\|\cdot\|$ is the Euclidean norm, $\boldsymbol{e} = -\nabla\mathcal{L}(\boldsymbol{x})/\|\nabla\mathcal{L}(\boldsymbol{x})\|$ and $\delta = \epsilon \|\nabla\mathcal{L}(\boldsymbol{x})\|/(d-1)$.

To derive the MH ratio, the key is to realize that the $A$ and $B$ updates are deterministic

$$q(\boldsymbol{z}'|\boldsymbol{z}) = \delta(\varphi(\boldsymbol{z}) - \boldsymbol{z}'), \tag{9}$$

where the transition map is generated by a dynamical system (Fang et al., 2014) for $\boldsymbol{z} = (\boldsymbol{x}, \boldsymbol{u})$,

$$\dot{\boldsymbol{z}}(t) = F(\boldsymbol{z}(t)). \tag{10}$$

Here, $\boldsymbol{z}(0) = \boldsymbol{z}$ and $z(T) = \varphi(\boldsymbol{z}) = \boldsymbol{z}'$. The drift vector field $F$ in canonical and microcanonical dynamics can be read from Equations (2) and (3), respectively. For the former, it equals $F_A(\boldsymbol{x}, \boldsymbol{u}) = (\boldsymbol{u}, 0)$ during the $A$ updates, and $F_B(\boldsymbol{x}, \boldsymbol{u}) = (0, -\nabla\mathcal{L}(\boldsymbol{x}))$ during the $B$ updates. For the latter, it equals $F_A(\boldsymbol{x}, \boldsymbol{u}) = (\boldsymbol{u}, 0)$ during the $A$ updates, and $F_B(\boldsymbol{x}, \boldsymbol{u}) = (0, -(1 - \boldsymbol{u}\boldsymbol{u}^T)\nabla\mathcal{L}(\boldsymbol{x})/(d-1))$ during the $B$ updates. These fields can be used to explicitly solve for the $A$ and $B$ updates; the solutions are given in Appendix B.3.

**Lemma 4.1.** *For proposals which are deterministic involutions generated by a dynamical system of the form* (10)*, $W$ in the MH acceptance probability* (1) *equals*

$$W(\boldsymbol{z}', \boldsymbol{z}) = -\log \frac{p(\boldsymbol{z}')}{p(\boldsymbol{z})} - \int_0^T \nabla \cdot F(\boldsymbol{z}(s)) ds.$$

*Proof.* The first term comes from the first factor in Equation (1). For the second term, observe that the ratio of transition probabilities is

$$\frac{q(\boldsymbol{z}|\boldsymbol{z}')}{q(\boldsymbol{z}'|\boldsymbol{z})} = \frac{\delta(\varphi(\boldsymbol{z}') - \boldsymbol{z})}{\delta(\varphi(\boldsymbol{z}) - \boldsymbol{z}')} = \frac{\delta(\varphi(\boldsymbol{z}') - \boldsymbol{z})}{\delta(\boldsymbol{z} - \varphi(\boldsymbol{z}'))} |\frac{\partial\varphi}{\partial\boldsymbol{z}}(\boldsymbol{z})| = |\frac{\partial\varphi}{\partial\boldsymbol{z}}(\boldsymbol{z})|,$$

where in the second step we have used reversibility, as well as standard properties of the delta function[5]. This last expression is the Jacobian determinant of the transition map $\varphi$. Finally, the second term of $W$ in Lemma 4.1 follows from Eq. (11) by the Abel–Jacobi–Liouville identity. $\square$

---

[4] Note that the MAMS dynamics of Equation (3) are not Hamiltonian, so for MAMS, $K$ is not kinetic energy in the standard sense. In Appendix B.5, a relationship between MAMS dynamics and a Hamiltonian dynamics for which $K$ actually is kinetic energy is given, justifying the name.

[5] Recall that $\delta(x-a)f(x) = \delta(x-a)f(a)$, and $\delta(f(x)) = \sum_i \delta(x-a_i)|\frac{df}{dx}a_i|^{-1}$, where $a_i$ are the roots of $f$. In our case, $f(z) = \phi(z) - z'$, so that $\delta(\phi(z) - z') = \delta(z - \phi(z'))|\frac{\partial\phi}{\partial z}(\phi(z'))|^{-1} = \delta(z - \phi(z'))|\frac{\partial\phi}{\partial z}(z)|^{-1}$

**Lemma 4.2.** *For a proposal of the form* $\varphi = \mathcal{T} \circ B_{\epsilon_N} \circ A_{\eta_N} \circ \dots B_{\epsilon_1} A_{\eta_1}$, *where* $\epsilon_k \in \mathbb{R}$ *and* $\eta_k \in \mathbb{R}$ *for all* $k$, *W in both HMC and MAMS equals the total energy change of the proposal, that is, the sum of all the energy changes:*

$$W(\varphi(\boldsymbol{z}), \boldsymbol{z}) = \sum_{k=1}^{N} V(A_{\epsilon_k}(\boldsymbol{z}_{k-1})) - V(\boldsymbol{z}_{k-1}) + K(B_{\epsilon_k} \circ A_{\epsilon_k}(\boldsymbol{z}_{k-1})) - K(A_{\epsilon_k}(\boldsymbol{z}_{k-1}))$$

*where* $\boldsymbol{z}_k = B_{\epsilon_k} \circ A_{\eta_k} \circ \dots B_{\epsilon_1} \circ A_{\eta_1}(\boldsymbol{z})$ *and energy changes are taken from Equations* (6) *and* (7) *for HMC, and* (6) *and* (8) *for MAMS.*

*Proof.* The position update $A$ in both HMC and MAMS has a vanishing divergence: $\nabla \cdot F_A = \frac{\partial u_i}{\partial x_i} = 0$, so during the position update, only the first term in Lemma 4.1 survives and $W(\boldsymbol{z}'\boldsymbol{z}) = -\log p(\boldsymbol{x}', \boldsymbol{u}')/p(\boldsymbol{x}, \boldsymbol{u}) = -\log p(\boldsymbol{x}')/p(\boldsymbol{x}) = \Delta V$ from Equation (6).

The velocity update in HMC has vanishing divergence $\nabla \cdot F_B = \frac{-\partial \nabla_i \mathcal{L}(\boldsymbol{x})}{\partial u_i} = 0$, so during the HMC velocity update, only the first term of $W$ in Lemma 4.1 survives and $W(\boldsymbol{z}', \boldsymbol{z}) = -\log p(\boldsymbol{x}', \boldsymbol{u}')/p(\boldsymbol{x}, \boldsymbol{u}) = -\log p(\boldsymbol{u}')/p(\boldsymbol{u}) = \Delta K$ from Equation (8).

The velocity update in MAMS, on the other hand, has a non-zero divergence:

$$\nabla \cdot F_B = -\|\nabla \mathcal{L}(\boldsymbol{x})\| \boldsymbol{u}(t) \cdot \boldsymbol{e} = \|\nabla \mathcal{L}(\boldsymbol{x})\| \frac{\sinh \delta + \cosh \delta (\boldsymbol{e} \cdot \boldsymbol{u})}{\cosh \delta + \sinh \delta (\boldsymbol{e} \cdot \boldsymbol{u})}$$

$$= -(d-1)\frac{d}{dt} \log\{\cosh \delta + \sinh \delta (\boldsymbol{e} \cdot \boldsymbol{u})\}.$$

In the first equality, we have used the divergence from Robnik and Seljak (2024), which we also re-derive in Appendix B.7. In the second equality, we used the explicit form of the velocity update from Appendix B.3, namely Equation (28). So we find that the velocity update for MAMS has

$$W(\varphi_{F_B}(\boldsymbol{z}), \boldsymbol{z}) = -\int_0^T \nabla \cdot F_B(\boldsymbol{z}(s))ds = (d-1)\log\{\cosh \delta + \boldsymbol{e} \cdot \boldsymbol{u} \sinh \delta\} = \Delta K$$

from Equation (7). A more direct derivation of the above is provided in Appendix B.6 for the interested reader.

$W$ of a composition of maps of the form in Lemma 4.1 is a sum of the individual $W$, i.e., $W(\varphi(\chi(\boldsymbol{z})), \boldsymbol{z}) = W(\varphi(\chi(\boldsymbol{z})), \chi(\boldsymbol{z})) + W(\chi(\boldsymbol{z}), \boldsymbol{z})$. This yields the formula in the statement of the theorem. $\qquad\square$

This result shows how to perform Metropolis adjustment in MAMS: analogously to HMC, for every $B$ or $A$ update that the algorithm performs in the proposal, the energy difference must be calculated and added to the cumulative energy difference $W$ ($B$ update energy difference is calculated by Equation (7), $A$ update by Equation (6)). Final MH acceptance probability is $\min(1, e^{-W})$. This is a favorable result, because both the HMC and MAMS numerical integrators keep the energy error small, even over long trajectories (Leimkuhler and Matthews, 2015), implying that a high acceptance rate can be maintained. As an empirical illustration that the MH acceptance probability from Lemma 4.2 is correct, Section 4 shows a histogram of 2 million samples from a 100-dimensional Gaussian (1st dimension shown) using the MH-adjusted kernel (orange), given a step size of 20. The kernel without MH adjustment is also shown (blue) and exhibits asymptotic bias.

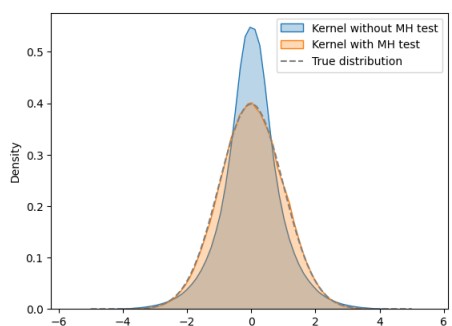

Figure 1: Histogram of MAMS samples with (orange) and without MH adjustment (blue) from 100-dim standard normal (1st dim shown). Step size is chosen very large ($\epsilon = 20$) to highlight the bias the MH step removes.

# 5 Sensitivity to hyperparameters

The performance of HMC is known to be very sensitive to the choice of the trajectory length, and the problem becomes even more pronounced for ill-conditioned targets, where different directions may require different trajectory lengths for optimal performance (Neal, 2011). This is further illustrated in Appendix D. Two solutions to this problem are randomizing the trajectory length (Bou-Rabee and Sanz-Serna, 2017) and replacing the full velocity refreshment with partial refreshment after every step, also known as the underdamped Langevin Monte Carlo (Horowitz, 1991b). Here, we will pursue both approaches with respect to MAMS.

**Random integration length**  We randomize the integration length by taking $n_k = \lceil 2h_k L/\epsilon \rceil$ integration steps to construct the $k$-th MH proposal. Here $h_k$ can either be random draws from the uniform distribution $\mathcal{U}(0, 1)$ or the $k$-th element of Halton's sequence, as recommended in Owen (2017); Hoffman et al. (2021). Other distributions of the trajectory length were also explored in the literature (Sountsov and Hoffman, 2022) but with no gain in performance. The factor of two is inserted to make sure that we do $L/\epsilon$ steps on average[6].

**Partial refreshment**  Partially refreshing the velocity after every step also has the effect of randomizing the time before the velocity coherence is lost, and therefore has similar benefits to randomizing the integration length (Jiang, 2023). However, while the flipping of velocity, needed for the deterministic part of the update to be an involution, is made redundant by a full resampling of velocity, this is not the case for partial refreshment. This results in rejected trajectories backtracking some of the progress that was made in the previously accepted proposals (Riou-Durand and Vogrinc, 2022). Skipping the velocity flip is possible, but it results in a small bias in the stationary distribution (Akhmatskaya et al., 2009), and it is not clear that it has any advantages over full refreshment. Two popular solutions for LMC are to either use a non-reversible MH acceptance probability as in Neal (2020); Hoffman and Sountsov (2022) or to add a full velocity refreshment before the MH step as in MALT (Riou-Durand and Vogrinc, 2022). We will prove that both can be straightforwardly used with microcanonical dynamics, and then concentrate on the MALT strategy in the remainder of the paper.

We will generate the Langevin dynamics by the "OBABO" scheme (Leimkuhler and Matthews, 2015), where BAB is the deterministic $\varphi_\epsilon$ map from Equation (5) and O is the partial velocity refreshment. In LMC, $O_\epsilon(\boldsymbol{u}) = c_1 \boldsymbol{u} + c_2 \boldsymbol{Z}$, where $Z$ is the standard normal distributed variable, $c_1 = e^{-\epsilon/L_{\text{partial}}}$ and $c_2 = \sqrt{1 - c_1^2}$. $L_{\text{partial}}$ is a parameter that controls the partial refreshment's strength and is comparable with HMC's trajectory length $L$. With microcanonical dynamics, a similar expression that additionally normalizes the velocity has been proposed (Robnik and Seljak, 2024):

$$O_\epsilon(\boldsymbol{u}) = \frac{c_1 \boldsymbol{u} + c_2 \boldsymbol{Z}/\sqrt{d}}{\left\| c_1 \boldsymbol{u} + c_2 \boldsymbol{Z}/\sqrt{d} \right\|}. \tag{11}$$

Denote by $\Delta(\boldsymbol{z}', \boldsymbol{z})$ the energy error accumulated in the *deterministic* ($\varphi_\epsilon$) part of the update. Note that for microcanonical Langevin dynamics, only the deterministic part of the update changes the energy, while in canonical Langevin dynamics, the $O$ update also does but is not included in $\Delta$.

**Theorem 5.1.** *The Metropolis-Hastings acceptance probability of the MAMS proposal $q(\boldsymbol{z}'|\boldsymbol{z})$, corresponding to $\mathcal{T}OBABO$ is $\min(1, e^{-\Delta(\boldsymbol{z}', \boldsymbol{z})})$.*

The generalized HMC strategy (Hoffman and Sountsov, 2022) only uses the one-step proposal, so Lemma 4.2 shows that it can be generalized to the microcanonical update, simply by using the microcanonical energy instead of the canonical energy. The MALT proposal, on the other hand, consists of $n$ LMC (or in our case, microcanonical LMC) steps and a full refreshment of the velocity, as shown in Algorithm 1.

**Theorem 5.2.** $\{\boldsymbol{x}_i\}_{i>0}$ *defined in Alg 1 is a Markov chain whose stationary distribution is $p(\boldsymbol{z})$.*

Proofs of both theorems are in Appendix A.

---

[6]More precisely, $\int_0^1 \lceil 2uL/\epsilon \rceil du \neq L/\epsilon$, because of the ceiling function. In the implementation, we use the correct expression, which is $n_k = \lceil 2yh_k L/\epsilon \rceil$, where $y = \frac{Y(Y+1)}{Y+1-L/\epsilon}$ and $Y = \lfloor 2L/\epsilon - 1 \rfloor$ is the integer part of $y$. This follows from solving $L/\epsilon = \mathbb{E}[n_k] = \frac{1+2+...+Y+(y-Y)(Y+1)}{y} = \frac{(Y+1)(y-Y/2)}{y}$ for $y$.

**Input:**
negative log-density function $\mathcal{L} : \mathbb{R}^d \to \mathbb{R}$, initial condition $\boldsymbol{x}^0 \in \mathbb{R}^d$, number of samples $N > 0$, step size $\epsilon > 0$, steps per sample $L/\epsilon \in \mathbb{N}$, partial refreshment parameter $L_{\text{partial}}$. The last three parameters can be determined automatically as in Section 6.

**Returns:** samples $\{\boldsymbol{x}^n\}_{n=1}^N$ from $p(\boldsymbol{x}) \propto e^{-\mathcal{L}(\boldsymbol{x})}$.

**for** $I \leftarrow 0$ **to** $N$ **do**
    $\boldsymbol{u} \sim \mathcal{U}_{S^{d-1}}$
    $\boldsymbol{z}^0 \leftarrow (\boldsymbol{x}^I, \boldsymbol{u})$
    $\delta \leftarrow 0$
    **for** $i \leftarrow 0$ **to** $n$ **do**
        $\boldsymbol{z} \leftarrow O_\epsilon(\boldsymbol{z}_i)$
        $\boldsymbol{z}' \leftarrow \Phi_\epsilon(\boldsymbol{z})$
        $\boldsymbol{z}^{i+1} \leftarrow O_\epsilon(\boldsymbol{z}')$
        $\delta \leftarrow \delta + \Delta(\boldsymbol{z}', \boldsymbol{z})$
    **end for**
    draw a random uniform variable $U \sim \mathcal{U}(0, 1)$
    **if** $U < e^{-\delta}$ **then**
        $\boldsymbol{x}^{I+1} \leftarrow \boldsymbol{z}^{n-1}[0]$
    **else**
        $\boldsymbol{x}^{I+1} \leftarrow \boldsymbol{z}^0[0]$
    **end if**
**end for**

**Algorithm 1:** MAMS - Langevin

## 6 Automatic hyperparameter tuning

MAMS has two hyperparameters, stepsize $\epsilon$ and the trajectory length $L$, where $L/\epsilon$ is the (average) number of steps in a proposal's trajectory. The Langevin version of the algorithm has an additional hyperparameter $L_{\text{partial}}$ that determines the partial refreshment strength during the proposal trajectories, i.e., the amount of Langevin noise. In addition, it is common to use a preconditioning matrix $M$ to linearly transform the configuration space, in order to reduce the condition number of the covariance matrix. The algorithm's performance crucially depends on these hyperparameters, so we here develop an automatic tuning scheme. First, the stepsize is tuned, then the preconditioning matrix, and finally, the trajectory length. $L_{\text{partial}}$ is directly set by the trajectory length, see Appendix D. We adopt a schedule similar to the one in (Robnik et al., 2024), where each of these three stages takes $10\%$ of the total sampling time, so that tuning does not significantly increase the total sampling cost.

**Stepsize** We extend the argument from (Beskos et al., 2013; Neal, 2011) to MAMS in Appendix C, showing that the optimal acceptance rate in MAMS is the same as in HMC, namely $65\%$. For HMC, larger acceptance rates have been observed to perform better in practice (Phan et al., 2019a), with some theoretical justification (Betancourt et al., 2015). We set the acceptance rate to $90\%$. In the first stage of tuning, we use a stochastic optimization scheme, dual averaging (Nesterov, 2009) from (Hoffman and Gelman, 2014), to adapt the step size until a desired acceptance rate is achieved.

**Preconditioning matrix** In the second stage, we determine the preconditioning matrix. A simple choice of diagonal preconditioning matrix is obtained by estimating variance along each parameter.

**Trajectory length** Microcanonical and canonical dynamics are extremely efficient in exploring the configuration space, while staying on the typical set. Therefore, we do not wish to reduce them to a comparatively inefficient diffusion process by adding too much noise, i.e., having too low $L$. On the other hand, we want to prevent the dynamics from being caught in cycles or quasi-cycles to maintain efficient exploration. Heuristically, we should send the dynamics in a new direction at the time scale that the dynamics needs to move to a different part of the configuration space, producing a new effective sample (Robnik et al., 2024). This suggests two approaches for tuning $L$ (Robnik et al., 2024).

The simpler is to estimate the size of the typical set by computing the average of the eigenvalues of the covariance matrix, which is equal to the mean of the variances in each dimension (Robnik et al.,

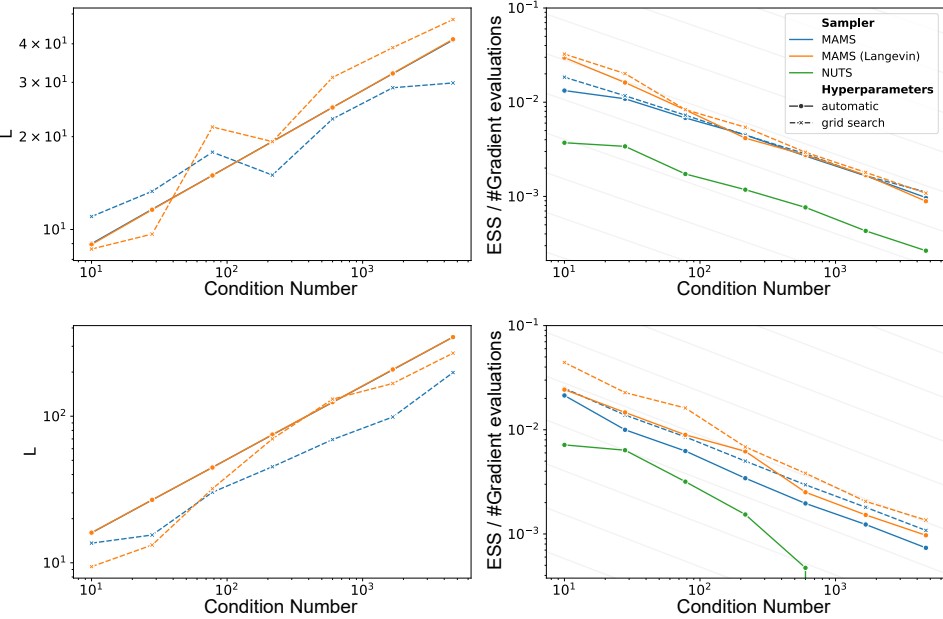

Figure 2: Tuning performance on Gaussians as a function of condition number. Gaussians are 100d with eigenvalues log-uniform distributed (top row) and outlier distributed (bottom). Left panels: the value of $L$ from the automatic tuning algorithm is shown with a solid line, and the optimal $L$ obtained by a grid search (dashed). As can be seen, automatic tuning achieves close to optimal values. Right panels: ESS (for the worst parameter) is shown as a function of condition number. MAMS tuning scheme (solid lines) achieves ESS, which is very close to the grid search results (dashed lines). MAMS scaling is similar to NUTS and goes as $ESS \propto \text{condition number}^{-1/2}$, which is shown as grey lines in the background. However, MAMS has around four times better proportionality constant.

2024). With a linearly preconditioned target, these variances are 1, and the estimate for the optimal $L$ is $L = \sqrt{d}$. We will use this as an initial value. A more refined approach is to set $L$ to be on the same scale as the time passed between effective samples:

$$L_{ALBA} \propto \mathbb{E}[\text{time between effective samples}] \quad = \mathbb{E}[\text{time between samples}] \, \tau_{\text{int}} = L \, \tau_{\text{int}}, \quad (12)$$

The proportionality constant is of order one and will be determined numerically, based on Gaussian targets. Integrated autocorrelation time $\tau_{\text{int}}$ is the ratio between the total number of (correlated) samples in the chain and the number of effectively uncorrelated samples. It depends on the observable $f(\boldsymbol{x})$ that we are interested in and can be calculated as

$$\tau_{\text{int}}[f] = 1 + 2 \sum_{t=1}^{\infty} \rho_t[f], \quad (13)$$

where

$$\rho_t[f] = \frac{\mathbb{E}[(f(\boldsymbol{x}(s)) - \mathbb{E}[f])(f(\boldsymbol{x}(s+t)) - \mathbb{E}[f])]}{\text{Var}[f]} \quad (14)$$

is the chain autocorrelation function in stationarity. We take $f(x_i) = x_i$ and harmonically average $\tau_{\text{int}}[x_i]$ over $i$. We determine the proportionality constant of Equation (12) in a way that $L$ equals the optimal L, determined by a grid search, for the standard Gaussian. We find a proportionality constant of 0.3 for MAMS without Langevin noise and 0.23 with Langevin noise.

## 7 Experiments

We aim to compare MAMS with NUTS, which is the main competitive *black-box* sampler in the single chain setting, see Section 2. We also compare against MALA (i.e., HMC with one step per trajectory), which is also popular in some settings.

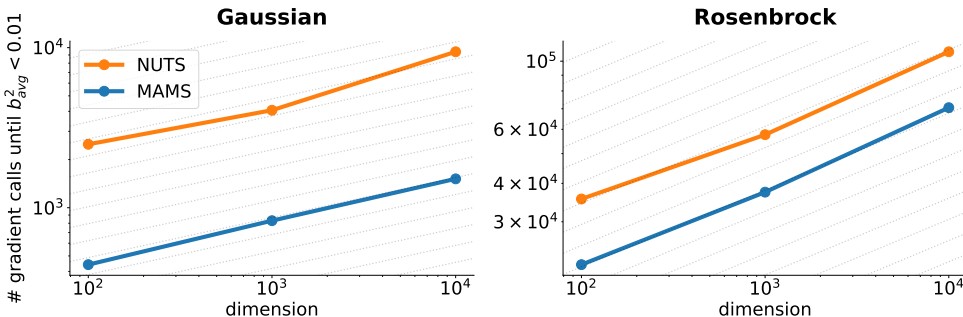

Figure 3: Sampling performance as a function of the dimensionality of the problem. On the left, the problem is a standard Gaussian, on the right, multiple independent copies of the Rosenbrock function (a banana-shaped target). Number of gradient calls to convergence scales as $d^{1/4}$ (grey lines in the background) for both MAMS and NUTS, but MAMS has a better proportionality constant.

**Evaluation metric** We follow Hoffman and Sountsov (2022) and define the squared error of the expectation value $\mathbb{E}[f(\boldsymbol{x})]$ as

$$b^2(f) = \frac{(\mathbb{E}_{\text{sampler}}[f] - \mathbb{E}[f])^2}{\text{Var}[f]}, \tag{15}$$

and consider the largest second-moment error across parameters, $b^2_{\max} \equiv \max_{1 \leq i \leq d} b^2(x_i^2)$, because in our problems of interest, there is typically a parameter of particular interest that has a significantly higher error than the other parameters (for example, a hierarchical parameter in many Bayesian models). For distributions which are a product of independent low-dimensional distributions (standard Gaussian, Rosenbrock function, and Cauchy problems), we take an average instead of the maximum because all parameters should have the same error. For the Cauchy distribution, the second moment $\mathbb{E}[x^2]$ diverges, so we instead consider the expected value of $-\log p(x)$, i.e., the entropy of the distribution. $b^2$ can be interpreted as an accuracy equivalent of 100 effective samples (Hoffman and Sountsov, 2022).

In typical applications, computing the gradients $\nabla \log p(\boldsymbol{x})$ dominates the total sampling cost, so we take the number of gradient evaluations as a proxy of wall-clock time. For very simple models, gradient calls might not dominate the cost, and the exact implementation of the numerical integration becomes important. Our implementation is efficient; for example, 1000 samples with L = 2 and stepsize = 1 for stochastic volatility take 1.5 seconds with MAMS on a single CPU and 4.8 seconds with NUTS. We do not report these numbers since they are irrelevant for the more expensive models that the method is meant to be applied to in practice. As in (Hoffman and Sountsov, 2022), we measure a sampler's performance as the number of gradient calls $n$ needed to achieve low error, $b^2_{\max} < 0.01$. We note that it is common to report effective sample size (ESS) per gradient evaluation instead, but both carry similar information, since $n$ can be interpreted as $100/(\text{ESS}/\# \text{ gradient evaluations})$. Furthermore, we argue that the former is of primary interest and the latter is only used as a proxy for the former.

**Scaling with the condition number and the dimensionality** Figure 2 compares MAMS with NUTS on 100-dimensional Gaussians with varying condition number. Two distributions of covariance matrix eigenvalues are tested: uniform in log and outlier distributed. *Outlier distributed* means that two eigenvalues are $\kappa$ while the other eigenvalues are 1. For both samplers, the number of gradients to low error scales with the condition number $\kappa$ as $\kappa^{-\frac{1}{2}}$, but MAMS is faster by a factor of around 4. Figure 3 compares MAMS and NUTS scaling with the problem's dimensionality. Both have the known $d^{\frac{1}{4}}$ scaling law (Neal, 2011), albeit MAMS has a better proportionality constant.

**Benchmarks** Table 1 compares MAMS with NUTS and MALA on a set of benchmark problems, mostly adapted from the Inference Gym (Sountsov et al., 2020). Problems vary in dimensionality (36–2429), are both synthetic and with real data, and include a distribution with a very long tail (Cauchy), a bimodal distribution (Bimodal), and many Bayesian inference problems. Problem details

|            | NUTS | MALA | MAMS | MAMS (Langevin) | MAMS (Grid Search) |
|------------|------|------|------|-----------------|--------------------|
| Gaussian | 19,652 | 11,010 | 3,249 | **3,172** | 3,121 |
| Banana | 95,519 | 140,524 | **14,078** | 14,818 | 15,288 |
| Bimodal | 210,758 | $> 10^6$ | 139,418 | **136,770** | 123,295 |
| Rosenbrock | 161,359 | $> 10^6$ | **94,184** | 103,545 | 93,782 |
| Cauchy | 171,373 | 824,429 | **110,404** | 155,963 | 87900 |
| Brownian | 29,816 | 597,119 | **13,528** | 15,232 | 14,015 |
| German Credit | 88,975 | $> 10^6$ | 55,748 | **49,979** | 52,265 |
| ItemResp | 76,043 | 249,470 | **45,371** | 56,902 | 45,640 |
| StochVol | 843,768 | $> 10^6$ | **430,088** | 510,190 | 431,957 |
| Funnel | $> 10^8$ | $> 10^7$ | 2,346,899 | **1,765,311** | 1,013,048 |

Table 1: Number of gradients calls needed to get the squared error on the worst second moment below 0.01. Lower is better; number of gradients is roughly proportional to wall clock time.

are in Appendix E.1. We do not include Bayesian neural networks because they lack an established ground truth (Izmailov et al., 2021), would require the use of stochastic gradients, which MAMS is not directly amenable to and because there is evidence that suggests that higher accuracy samples are not necessary for the good performance (Wenzel et al., 2020), so unadjusted methods achieve superior performance (Sommer et al., 2024).

For all algorithms, we use an initial run to find a diagonal preconditioning matrix. For NUTS and MALA, the only remaining parameter to tune is step size, which is tuned by dual averaging, targeting $80\%$ acceptance rate for NUTS and $57.4\%$ for MALA (Neal, 2011). For MAMS, we further tune $L$ using the scheme of Section 6. We take the adaptation steps as our burn-in, initializing the chain with the final state returned by the adaptation procedure. NUTS is run using the BlackJax (Cabezas et al., 2024) implementation, with the provided window adaptation scheme. Table 1 shows the number of gradient calls in the chain (excluding tuning) used to reach squared error of $0.01$. To reduce variance in these results, we run at least 128 chains for each problem and take the median of the error across chains at each step.

**Results**   In all cases, MAMS outperforms NUTS, typically by a factor of two at worst and seven at best. MALA drastically underperforms relative to MAMS and NUTS. Using Langevin noise instead of the trajectory length randomization has little effect in most cases, analogous to the situation for HMC (Jiang, 2023; Riou-Durand et al., 2023). For Neal's Funnel, we need an acceptance rate of 0.99 for MAMS to converge. We were unable to obtain convergence for NUTS. This problem is a known NUTS failure mode, so it is of note that MAMS converges.

To assess how successful the tuning scheme from Section 6 is at finding the optimal value of $L$, we perform a grid search over $L$, by first performing a long NUTS run to obtain a diagonal of the covariance matrix and an initial $L$, and then for each new candidate value of $L$, tuning step size by dual averaging with a target acceptance rate of $0.9$. In Table 1 we show the number of gradients to low error using this optimal $L$. Performance is very close to optimal on all benchmark problems.

## 8    Conclusions

Our core contribution is MAMS, an out-of-the-box gradient-based sampler applicable in the same settings as NUTS HMC and intended as a successor to it. Our experiments found substantial performance gains for MAMS over NUTS in terms of statistical efficiency. These experiments were on problems varying in dimension (up to $10^4$) and included real datasets, multimodality, and long tails. This said, to reach the maturity of NUTS, the method needs to be battle-tested over many years on an even broader variety of problems.

We note that MAMS is simple to implement, with little code change from standard HMC. A promising future direction is the many-short-chains MCMC regime (Hoffman and Sountsov, 2022; Margossian et al., 2024), since MAMS is not as control-flow heavy as NUTS and since MAMS with Langevin noise can have a fixed number of steps per trajectory, making parallelization more efficient (Sountsov et al., 2024).

## Acknowledgments and Disclosure of Funding

This material is based upon work supported in part by the Heising-Simons Foundation grant 2021-3282 and by NSF CSSI grant award number 2311559.

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

# A   Metropolis adjusted Microcanonical Langevin dynamics proofs

Denote by $o(z'|z)$ the density corresponding to the $O$ update and by $q(z'|z)$, the density corresponding to the single step proposal $\mathcal{T}OBABO$. We will use a shorthand notation for the time reversal: $\overline{z} = \mathcal{T}(z)$ and denote by $\Delta(z', z)$ the energy error accumulated in the *deterministic* part of the update.

## A.1   Proof of Theorem 5.1

*Proof.* For the MH ratio we will need

$$\frac{q(z|\overline{z}')}{q(\overline{z}'|z)} = \frac{\int o(\overline{z}|Z')\,\delta(Z', \varphi(Z))\,o(Z|\overline{z}')dZdZ'}{\int o(z'|Z')\,\delta(Z', \varphi(Z))\,o(Z|z)dZdZ'} = \frac{\int o(\overline{z}|\varphi(Z))\,o(Z|\overline{z}')dZ}{\int o(z'|\varphi(Z))\,o(Z|z)dZ}, \quad (16)$$

where we have used the delta function to evaluate the integral over $Z'$.

We can further simplify the numerator

$$\int o(\overline{z}|\varphi(Z))\,o(Z|\overline{z}')dZ = \int o(z|\overline{\varphi(Z)})\,o(\overline{Z}|z')dZ = \int o(z|\overline{\varphi(\overline{Z})})\,o(Z|z')dZ$$

$$= \int o(z|\varphi^{-1}(Z))\,o(Z|z')dZ = \int o(z|Z)\,o(\varphi(Z)|z')\left|\frac{\partial\varphi(Z)}{\partial Z}\right|dZ = \int o(z'|\varphi(Z))\,o(Z|z)\left|\frac{\partial\varphi(Z)}{\partial Z}\right|dZ.$$

In the first step we have used that $o(\overline{x}|\overline{y}) = o(x|y)$ and that time reversal is an involution. In the second step, we have performed a change of variables from $\overline{Z}$ to $Z$ (for which, the Jacobian determinant of the transformation is 1). In the third step we used that $\overline{\varphi(\overline{Z})} = \varphi^{-1}(Z)$. In the fourth step we change variables to $\varphi^{-1}(Z)$ instead of $Z$. In the last step we use that $o(y|x) = o(x|y)$.

Since $o$ only connects states with the same $x$ there is only one $Z$ which makes the integral nonvanishing and we get

$$\frac{q(\overline{z}'|z)}{q(z|\overline{z}')} = \left|\frac{\partial\varphi(Z)}{\partial Z}\right|,$$

as if there were no O updates. The O updates also preserve the target density, so we see that the acceptance probability is only concerned with the BAB part of the update. In this case, the desired acceptance probability was already derived in Lemma 4.2. $\qquad\square$

## A.2   Proof of Theorem 5.2

*Proof.* Following a similar structure of the proof as in (Riou-Durand and Vogrinc, 2022) we will work on the space of trajectories $z_{0:L} = (z_0, \ldots, z_L) \in \mathcal{M}^{L+1}$. We will define a kernel $\mathcal{Q}$ on the space of trajectories, with $q$ as a marginal-$x_0$ kernel. We will prove that $\mathcal{Q}$ is reversible with respect to the extended density

$$\mathcal{P}(z_{0:L}) = \prod_{i=1}^{L} q(z_i|z_{i-1})p(z_0),$$

and use it to show that $q$ is reversible with respect to the marginal $p(x_0)$.

We define the Gibbs update, corresponding to the conditional distribution $\mathcal{P}(\cdot|x_0)$:

$$\mathcal{G}(z'_{0:L}|z_{0:L}) = \delta(x'_0 - x_0)U_{S^{d-1}}(u'_0)\prod_{i=1}^{L} q(z'_i|z'_{i-1}).$$

The Gibbs kernel $\mathcal{G}$ is reversible with respect to $\mathcal{P}$ by construction. Built upon a deterministic proposal of the backward trajectory

$$\overline{z}_{0:L} = (\overline{z}_L, \overline{z}_{L-1}, \ldots, \overline{z}_0), \quad (17)$$

we introduce a Metropolis update:

$$M(z'_{0:L}|z_{0:L}) = P_{MH}\,\delta(z'_{0:L} - \overline{z}_{0:L}) + (1 - P_{MH})\delta(z'_{0:L} - z_{0:L}),$$

where $P_{MH}(\boldsymbol{z}_{0:L}) = \min(1, e^{-\Delta(\boldsymbol{z}_{0:L})})$. For $\eta > 0$, the distribution $\mathcal{P}$ admits a density with respect to Lebesgue's measure. Therefore

$$e^{-\Delta(\boldsymbol{z}_{0:L})} = \frac{\mathcal{P}(\overline{\boldsymbol{z}}_{0:L})}{\mathcal{P}(\boldsymbol{z}_{0:L})}\left|\frac{\partial \overline{\boldsymbol{z}}_{0:L}}{\partial \boldsymbol{z}_{0:L}}\right| \tag{18}$$

ensures that the Metropolis kernel $M$ is reversible with respect to $\mathcal{P}$.

Before proceeding with the proof, we express Equation (18) in a simple, easy-to-compute form. The Jacobian is $\frac{\partial \overline{\boldsymbol{z}}_{0:L}}{\partial \boldsymbol{z}_{0:L}} = \sigma \otimes \frac{\partial \overline{\boldsymbol{z}}}{\partial \boldsymbol{z}}$, where $\sigma$ is the matrix of the permutation $\sigma(i) = L - i$ and $\frac{\partial \overline{\boldsymbol{z}}}{\partial \boldsymbol{z}} = I_{d \times d} \oplus -I_{d-1 \times d-1}$. Both of these matrices have determinant $\pm 1$, so the determinant of their Kronecker product is also $\pm 1$ and its absolute value is 1.

We get

$$e^{-\Delta(\boldsymbol{z}_{0:L})} = \frac{\mathcal{P}(\overline{\boldsymbol{z}}_{0:L})}{\mathcal{P}(\boldsymbol{z}_{0:L})} = \frac{p(\overline{\boldsymbol{z}}_L)\prod_{i=1}^{L} q(\overline{\boldsymbol{z}}_{i-1}|\overline{\boldsymbol{z}}_i)}{p(\boldsymbol{z}_0)\prod_{i=1}^{L} q(\boldsymbol{z}_i|\boldsymbol{z}_{i-1})} = \frac{\prod_{i=1}^{L} p(\overline{\boldsymbol{z}}_i)\prod_{i=1}^{L} q(\overline{\boldsymbol{z}}_{i-1}|\overline{\boldsymbol{z}}_i)}{\prod_{i=1}^{L} p(\boldsymbol{z}_{i-1})\prod_{i=1}^{L} q(\boldsymbol{z}_i|\boldsymbol{z}_{i-1})} \tag{19}$$

$$= \prod_{i=1}^{L} \frac{q(\overline{\boldsymbol{z}}_{i-1}|\overline{\boldsymbol{z}}_i)p(\overline{\boldsymbol{z}}_i)}{q(\boldsymbol{z}_i|\boldsymbol{z}_{i-1})p(\boldsymbol{z}_{i-1})} = e^{-\sum_{i=1}^{L} \Delta(\boldsymbol{z}_i, \boldsymbol{z}_{i-1})},$$

where $\Delta(\boldsymbol{z}_i, \boldsymbol{z}_{i-1})$ is the energy error in step $i$, by Theorem 5.1.

We are now in a position to define the trajectory-space kernel:

$$\mathcal{Q} = \mathcal{G}M\mathcal{G}. \tag{20}$$

The palindromic structure of $\mathcal{Q}$ ensures reversibility with respect to $\mathcal{P}$. Since the transition $\mathcal{G}(\cdot|\boldsymbol{z}_{0:L}) = \mathcal{G}(\cdot|\boldsymbol{x}_0)$ only depends on the starting position $\boldsymbol{x}_0 \in \mathbb{R}^d$ and $p(\boldsymbol{x})$ is the marginal of $\mathcal{P}$, we obtain that $q(\boldsymbol{x}_0'|\boldsymbol{x}_0) = \int \mathcal{Q}(\boldsymbol{z}_{0:L}'|\boldsymbol{z}_{0:L})d\boldsymbol{u}_0 \prod_{i=1}^{L} d\boldsymbol{z}_i$ defines marginally a Markov kernel on $\mathbb{R}^d$, reversible with respect to $p$. In particular, the distribution of $\{\boldsymbol{x}_i\}_{i \geq 0}$ in Algorithm 1 coincides with the distribution of a Markov chain generated by $q$. $\qquad\square$

# B  Microcanonical dynamics

In this appendix, we establish a relationship between the microcanonical dynamics of Equation (3) and a Hamiltonian system with energy $E$ from which it can be derived by a time-rescaling operation. As well as motivating the dynamics of Equation (3), this allows us to show that $W$ in Lemma 4.2 for the dynamics of Equation (3) corresponds to the change in energy $E$ of the Hamiltonian system. We also provide a complete derivation of the form of $W$ for microcanonical dynamics. Familiarity with the basics of Hamiltonian mechanics is assumed throughout.

## B.1  Sundman transformation

We begin by introducing a transformation to a Hamiltonian system known as a Sundman transform (Leimkuhler and Reich, 2004)

$$S(F)(\boldsymbol{z}(t)) = w(\boldsymbol{z}(t))F(\boldsymbol{z}(t)),$$

where $w$ is any function $\mathbb{R}^{2d} \to \mathbb{R}$. Intuitively, this is a $\boldsymbol{z}$-dependent time rescaling of the dynamics. Therefore it is not surprising that:

**Lemma B.1.** *The integral curves of $S(F)$ are the same as of $F$ (Skeel, 2009)*

*Proof.* To see this, first use $\boldsymbol{z}_G$ to refer to the dynamics from a field $G$, and posit that $\boldsymbol{z}_{S(F)}(s) = \boldsymbol{z}_F(t(s))$, where $\frac{dt(s)}{ds} = w(\boldsymbol{z}(s))$. Then we see that

$$\frac{d\boldsymbol{z}_{S(F)}(s)}{dt} = \frac{d\boldsymbol{z}_F(t(s))}{ds} = \frac{d\boldsymbol{z}_F(t)}{dt}\frac{dt}{ds} = F(\boldsymbol{z}_F(s))w(\boldsymbol{z}_F(s)),$$

which shows that, indeed, $\boldsymbol{z}_{S(F)} = \boldsymbol{z}_F \circ s$, where $s$ is a function $\mathbb{R} \to \mathbb{R}$, which amounts to what we set out to show.

$\square$

However, note that the stationary distribution is not necessarily preserved, on account of the phase space dependence of the time-rescaling, which means that in a volume of phase space, different particles will move at different velocities.

### B.2  Obtaining the dynamics of Equation (3)

Consider the Hamiltonian system[7] given by $H = T + V$, with $T(\boldsymbol{\Pi}) = (d-1)\log\|\boldsymbol{\Pi}\|$ and $V(\boldsymbol{x}) = \mathcal{L}(\boldsymbol{x})$. Here, $\boldsymbol{\Pi}$ is the canonical momentum associated to position $\boldsymbol{x}$. Then the dynamics derived from Hamilton's equations of motion are:

$$\frac{d}{dt}\begin{bmatrix}\boldsymbol{x}\\\boldsymbol{\Pi}\end{bmatrix} = \begin{bmatrix}\frac{\partial H}{\partial \boldsymbol{\Pi}}\\-\frac{\partial H}{\partial \boldsymbol{x}}\end{bmatrix} = \begin{bmatrix}(d-1)\frac{\boldsymbol{\Pi}}{\|\boldsymbol{\Pi}\|^2}\\-\nabla_{\boldsymbol{x}}\mathcal{L}(\boldsymbol{x})\end{bmatrix} := F(\boldsymbol{z}). \tag{21}$$

Any Hamiltonian dynamics has $p(\boldsymbol{z}) \propto \delta(H - C)$ as a stationary distribution, which can be sampled from by integrating the equations if ergodicity holds. As observed in Steeg and Galstyan (2021) and Robnik et al. (2024), the closely related Hamiltonian $d\log\|\boldsymbol{\Pi}\| + \mathcal{L}(\boldsymbol{x})$ has the property that the marginal of this stationary distribution is the desired target, namely $p(\boldsymbol{x}) \propto e^{-\mathcal{L}(\boldsymbol{x})}$. However, numerical integration of these equations is unstable due to the $\frac{1}{\|\boldsymbol{\Pi}\|^2}$ factor, and moreover, MH adjustment is not possible since numerical integration induces error in $H$, which would result in proposals always being rejected, due to the delta function.

Both problems can be addressed with a Sundman transform and a subsequent change of variables. To that end, we choose $w(\boldsymbol{z}) = \|\boldsymbol{\Pi}\|/(d-1)$ (which corresponds, up to a factor, to the weight $r$ in Steeg and Galstyan (2021), and to $w$ in Robnik et al. (2024)), we obtain:

$$\frac{d}{dt}\begin{bmatrix}\boldsymbol{x}\\\boldsymbol{\Pi}\end{bmatrix} = \begin{bmatrix}\boldsymbol{\Pi}/\|\boldsymbol{\Pi}\|\\-\nabla\mathcal{L}(\boldsymbol{x})\|\boldsymbol{\Pi}\|/(d-1)\end{bmatrix}. \tag{22}$$

Changing variables to $\boldsymbol{u} = \boldsymbol{\Pi}/\|\boldsymbol{\Pi}\|$, we obtain precisely the microcanonical dynamics of Equation (3):

$$\frac{d}{dt}\begin{bmatrix}\boldsymbol{x}\\\boldsymbol{u}\end{bmatrix} = \begin{bmatrix}\boldsymbol{u}\\-(I - \boldsymbol{u}\boldsymbol{u}^T)\nabla\mathcal{L}(\boldsymbol{x})/(d-1)\end{bmatrix} := \begin{bmatrix}B_{\boldsymbol{x}}\\B_{\boldsymbol{u}}\end{bmatrix},$$

where we have used that the Jacobian $\frac{d\boldsymbol{u}}{d\boldsymbol{\Pi}} = \frac{1}{\|\boldsymbol{\Pi}\|}(I - \frac{\boldsymbol{\Pi}\boldsymbol{\Pi}^T}{\|\boldsymbol{\Pi}\|^2})$. Note that $B_{\boldsymbol{x}} = S(F)_{\boldsymbol{x}}$, since this final change of variable only targets $\boldsymbol{\Pi}$.

### B.3  Discrete updates

For completeness, we here state the position and velocity updates of the canonical and microcanonical dynamics, which are obtained by solving dynamics at fixed velocity for the position update and at fixed position for the velocity update. For canonical dynamics, this amounts to solving

$$\frac{d}{d\epsilon}A_\epsilon = \boldsymbol{u}(t) \qquad \frac{d}{d\epsilon}B_\epsilon = -\nabla\mathcal{L}(\boldsymbol{x}(t)), \tag{23}$$

with initial condition $A_0 = \boldsymbol{x}(t)$ and $B_0 = \boldsymbol{u}(t)$. These solution is trivial:

$$A_\epsilon = \boldsymbol{x}(t) + \epsilon\boldsymbol{u}(t) \qquad B_\epsilon = \boldsymbol{u}(t) - \epsilon\nabla\mathcal{L}(\boldsymbol{x}(t)). \tag{24}$$

---

[7]Here we follow Robnik et al. (2024) and Steeg and Galstyan (2021), but our Hamiltonian differs by a factor, to avoid the need for a weighting scheme used in those papers.

For microcanonical dynamics, one needs to solve

$$\frac{d}{d\epsilon} A_\epsilon = \boldsymbol{u}(t) \qquad \frac{d}{d\epsilon} B_\epsilon = -(1 - \boldsymbol{u}(t)\boldsymbol{u}(t)^T)\nabla\mathcal{L}(\boldsymbol{x}(t))/(d-1), \tag{25}$$

with initial condition $A_0 = \boldsymbol{x}(t)$ and $B_0 = \boldsymbol{u}(t)$. The velocity equation is a vector version of the Riccati equation (Steeg and Galstyan, 2021). Denote $\boldsymbol{g} = -\nabla\mathcal{L}(\boldsymbol{x}(t))/(d-1)$ and replace the variable $B_\epsilon$ by $\boldsymbol{y}_\epsilon$, such that

$$B_\epsilon = \frac{\frac{d}{d\epsilon}\boldsymbol{y}_\epsilon}{\boldsymbol{g} \cdot \boldsymbol{y}_\epsilon}. \tag{26}$$

This is convenient, because the equation for $B_\epsilon$ is a nonlinear first-order differential equation, but the equation for $\boldsymbol{y}_\epsilon$ is a *linear* second-order differential equation

$$\frac{d^2}{d\epsilon^2}\boldsymbol{y}_\epsilon = (\boldsymbol{g}\boldsymbol{g}^T)\boldsymbol{y}_\epsilon, \tag{27}$$

which is easy to solve and yields the updates

$$A_\epsilon = \boldsymbol{x}(t) + \epsilon\boldsymbol{u}(t) \qquad B_\epsilon = \frac{\boldsymbol{u}(t) + (\sinh\delta + \boldsymbol{e}\cdot\boldsymbol{u}(t)(\cosh\delta - 1))\boldsymbol{e}}{\cosh\delta + \boldsymbol{e}\cdot\boldsymbol{u}(t)\sinh\delta}, \tag{28}$$

where $\delta = \epsilon\left\|\nabla\mathcal{L}(\boldsymbol{x}(t))\right\|/(d-1)$ and $\boldsymbol{e} = -\nabla\mathcal{L}(\boldsymbol{x})/\left\|\nabla\mathcal{L}(\boldsymbol{x})\right\|$.

### B.4  Obtaining the stationary distribution of Equation (3)

We can derive the stationary distribution of Equation (3) following the approach of Tuckerman (2023). There, it is shown that for a flow $F$, if there is a $g$ such that $\frac{d}{dt}\log g = -\nabla \cdot F$, and $\Lambda$ is the conserved quantity under the dynamics, then $p(\boldsymbol{z}) \propto g(\boldsymbol{z})f(\Lambda(\boldsymbol{z}))$, where $f$ is any function.

We note that $\nabla \cdot F = \boldsymbol{u} \cdot \nabla\mathcal{L}(x) = \frac{d}{dt}\mathcal{L}(\boldsymbol{x})$, using Appendix B.7 in the first step. Therefore $\log g = -\mathcal{L}(\boldsymbol{x})$. Further, $|\boldsymbol{u}|$ is preserved by the dynamics if we initialize with $|\boldsymbol{u}_0| = 1$, as can easily be seen: $\frac{d}{dt}(\boldsymbol{u}\cdot\boldsymbol{u}) = 2\boldsymbol{u}\cdot\dot{\boldsymbol{u}} = 2\boldsymbol{u}\cdot(I-\boldsymbol{u}\boldsymbol{u}^T)(-\nabla\mathcal{L}(\boldsymbol{x})/(d-1)) = 2(1-\boldsymbol{u}\cdot\boldsymbol{u})(\boldsymbol{u}\cdot-\nabla\mathcal{L}/(d-1)) = 0$. Thus a stationary distribution is:

$$p(\boldsymbol{x}, \boldsymbol{u}) \propto e^{-\mathcal{L}(\boldsymbol{x})}\delta(\|u\| - 1). \tag{29}$$

Importantly, because even the discretized dynamics are norm preserving, the condition $\delta(|\boldsymbol{u}| - 1)$ is always satisfied, so that $\frac{p(\boldsymbol{z}')}{p(\boldsymbol{z})}$ is always well defined. This makes it possible to perform MH adjustment, in contrast to the original Hamiltonian dynamics as discussed in Appendix B.2.

### B.5  $W$ as energy change

In the non-equilibrium physics literature, $W$ (termed the dissipation function) is interpreted as work done on the system and the second term in Lemma 4.1 is the dissipated heat (Evans and Searles, 1994, 2002; Sevick et al., 2008). $W$ plays a central role in fluctuation theorems, for example, Crook's relation (Crooks, 1999) states that the transitions $\boldsymbol{z} \to \boldsymbol{z}'$ are more probable than $\boldsymbol{z}' \to \boldsymbol{z}$ by a factor $e^{W(\boldsymbol{z}',\boldsymbol{z})}$. In statistics, this fact is used by the MH algorithm to obtain reversibility, or *detailed balance*, a sufficient condition for convergence to the target distribution.

Here we will justify why it can also be interpreted as an energy change in microcanonical dynamics.

**Lemma B.2.** *$W$, calculated for the microcanonical dynamics over a time interval $[0, T]$ is equal to $\Delta E$ of the Hamiltonian $(d-1)\log\|\boldsymbol{\Pi}\| + \mathcal{L}(\boldsymbol{x})$ for an interval $[s(0), s(T)]$, where $s$ is the time rescaling arising from the Sundman transformation $w(\boldsymbol{z}) = |\boldsymbol{\Pi}_F|/(d-1)$.*

*Proof.* Recall that for a flow field $F$:

$$W(\boldsymbol{z}_F(T), \boldsymbol{z}_F(0)) = -\log\frac{p(\boldsymbol{z}_F(T))}{p(\boldsymbol{z}_F(0))} - \int_0^T \nabla \cdot F(\boldsymbol{z}_F(s))ds. \tag{30}$$

Given the form of the stationary distribution induced by $B$, derived in Appendix B.4, we see that the first term of the work, $\log \frac{P(\boldsymbol{z}_B(0))}{P(\boldsymbol{z}_B(T))} = \mathcal{L}(\boldsymbol{x}_B(T)) - \mathcal{L}(\boldsymbol{x}_B(0)) = \mathcal{L}(\boldsymbol{x}_{S(F)}(T)) - \mathcal{L}(\boldsymbol{x}_{S(F)}(0)) = \mathcal{L}(\boldsymbol{x}_F(s(T))) - \mathcal{L}(\boldsymbol{x}_F(s(0)))$ which is equal to $\Delta V$ for an interval of time $[s(0), s(T)]$.

As for the second term, observe that

$$\frac{dK(\boldsymbol{\Pi}(t(s)))}{ds} = \frac{\partial H}{\partial \boldsymbol{\Pi}} \cdot \frac{d\boldsymbol{\Pi}}{dt} \frac{dt}{ds} = \frac{d\boldsymbol{x}}{dt} \frac{dt}{ds} \cdot \frac{d\boldsymbol{\Pi}}{dt} = \boldsymbol{u} \cdot (-\nabla \mathcal{L}(\boldsymbol{x})) = -\nabla \cdot B,$$

which is precisely the integrand of the second term.

$\square$

This shows that $W = \Delta K + \Delta V = \Delta E$, where $\Delta E$ is the energy change of the original Hamiltonian, over the rescaled time interval $[s(0), s(T)]$. As we know, $\Delta E = 0$ for the exact Hamiltonian flow, and indeed $W = 0$ for the exact dynamics of Equation (3), which is to say that for the exact dynamics, no MH correction would be needed for an asymptotically unbiased sampler.

However, our practical interest is in the discretized dynamics arising from a Velocity Verlet numerical integrator. In this case, we wish to calculate $W$ for $B_{\boldsymbol{u}}$ and $B_{\boldsymbol{x}}$ separately, and consider the sum, noting that $W$ is an additive quantity with respect to the concatenation of two dynamics. Considering $W$ with respect to only $B_{\boldsymbol{x}}$, we see that the first term of $W$ remains $\Delta V$, since the stationary distribution gives uniform weight to all values of $\boldsymbol{u}$ of unit norm, and the dynamics are norm preserving. The second term vanishes, because $\nabla_{\boldsymbol{x}} B_{\boldsymbol{x}} = \nabla_{\boldsymbol{x}} \boldsymbol{u} = 0$. As for $B_{\boldsymbol{u}}$, since the norm preserving change in $\boldsymbol{u}$ leaves the density unchanged, the first term of $W$ vanishes. Meanwhile, the second term is $\Delta K$, from the above derivation, since $\nabla \cdot B = \nabla_{\boldsymbol{x}} \cdot B_{\boldsymbol{x}} + \nabla_{\boldsymbol{u}} \cdot B_{\boldsymbol{u}} = \nabla_{\boldsymbol{u}} \cdot B_{\boldsymbol{u}}$. Thus, the full $W$ is equal to $\Delta V + \Delta K = \Delta E$, as desired. For HMC, it is easily seen that $W$ for $F_{\boldsymbol{x}}$ is $\Delta V$, and for $F_{\boldsymbol{u}}$ is $\Delta T$. Putting this together, we maintain the result of Lemma B.2, but now in a setting where $W$ is not 0 so that MH adjustment is of use.

### B.6 Direct calculation of velocity update $W$

We here provide a self-contained derivation of the MH ratio for the velocity update from Equation (28). The MH ratio is a scalar with respect to state space transformations, i.e. it is the same in all coordinate systems. We can therefore select convenient coordinates for its computation. We will choose spherical coordinates in which $\boldsymbol{e}$ is the north pole and

$$\boldsymbol{u} = \cos \vartheta \boldsymbol{e} + \sin \vartheta \boldsymbol{f}, \tag{31}$$

for some unit vector $\boldsymbol{f}$, orthogonal to $\boldsymbol{e}$. $\vartheta$ is then a coordinate on the $S^{d-1}$ manifold. The velocity updating map from Equation (28),

$$\boldsymbol{u}' = \frac{1}{\cosh \delta + \cos \vartheta \sinh \delta} \boldsymbol{u} + \frac{\sinh \delta + \cos \vartheta (\cosh \delta - 1)}{\cosh \delta + \cos \vartheta \sinh \delta} \boldsymbol{e} \tag{32}$$

$$= \frac{\sinh \delta + \cos \vartheta \cosh \delta}{\cosh \delta + \cos \vartheta \sinh \delta} \boldsymbol{e} + \frac{\sin \vartheta}{\cosh \delta + \cos \vartheta \sinh \delta} \boldsymbol{f}, \tag{33}$$

can be expressed in terms of the $\vartheta$ variable:

$$\cos \vartheta' = \frac{\sinh \delta + \cos \vartheta \cosh \delta}{\cosh \delta + \cos \vartheta \sinh \delta} \qquad \sin \vartheta' = \frac{\sin \vartheta}{\cosh \delta + \cos \vartheta \sinh \delta}. \tag{34}$$

The Jacobian of the $\vartheta \mapsto \vartheta'$ transformation is

$$\left| \frac{d\vartheta'}{d\vartheta} \right| = \left| \frac{d\vartheta'}{d\cos \vartheta'} \frac{d\cos \vartheta'}{d\vartheta} \right| = \frac{1}{|\cosh \delta + \cos \vartheta \sinh \delta|} \tag{35}$$

and the density ratio is

$$\frac{p(\vartheta')}{p(\vartheta)} = \sqrt{\frac{g(\vartheta')}{g(\vartheta)}} = \left( \frac{\sin \vartheta'}{\sin \vartheta} \right)^{d-2} = \frac{1}{(\cosh \delta + \cos \vartheta \sinh \delta)^{d-2}}, \tag{36}$$

where $g$ is the metric determinant on a $S^{d-1}$ sphere. Combining the two together yields

$$W = (d-1) \log \left( \cosh \delta + \cos \vartheta \sinh \delta \right), \tag{37}$$

which is the kinetic energy from Equation (8).

### B.7 Direct calculation of the velocity update divergence

For completeness, we here derive the divergence of the microcanonical velocity update flow field $F$. We will use the divergence theorem, which states that the integral of the divergence of a vector field over some volume $\Omega$ equals the flux of this vector field over the boundary of $\Omega$. Here, flux is $F \cdot n$ where $n$ is the unit vector, normal to the boundary.

We will use the coordinate system defined in Equation (31) and pick as the volume $\Omega$ a thin spherical shell, centered around the north pole $e$ and spanning the $\vartheta$ range $[\vartheta, \vartheta + \Delta\vartheta]$. The boundary of $\Omega$ are two spheres in $d - 2$ dimensions with radia $\sin \vartheta$ and $\sin(\vartheta + \Delta\vartheta)$. Note that $F$ is normal to this boundary and flux is a constant on each shell. It is outflowing on the boundary which is closer to the north pole and inflowing on the other boundary.

Note that for $\Delta\vartheta \to 0$, we have that $\nabla \cdot F$ is a constant on $\Omega$. The divergence theorem in this limit therefore implies

$$(\nabla \cdot F)\, V(S^{d-2})(\sin \vartheta)^{d-2} = -\frac{d}{d\vartheta}\big(|F|V(S^{d-2})(\sin \vartheta)^{d-2}\big), \tag{38}$$

where $V(S^{d-2})$ is the volume of the unit sphere in d-2 dimensions and we have used that the volume of a $n$-dimensional sphere with radius $r$ is $V(S^n)r^n$. By rearranging we get:

$$\nabla \cdot F = -\frac{\frac{d}{d\vartheta}\big(|F|(\sin \vartheta)^{d-2}\big)}{(\sin \vartheta)^{d-2}}. \tag{39}$$

We have

$$F = \frac{\|\nabla\mathcal{L}(\boldsymbol{x})\|}{d-1}(1 - \boldsymbol{u}\boldsymbol{u}^T)\boldsymbol{e}, \tag{40}$$

so

$$|F| = \frac{\|\nabla\mathcal{L}(\boldsymbol{x})\|}{d-1}\sqrt{\boldsymbol{e}(1 - \boldsymbol{u}\boldsymbol{u}^T)\boldsymbol{e}} = \frac{\|\nabla\mathcal{L}(\boldsymbol{x})\|}{d-1}\sqrt{1 - (\boldsymbol{e} \cdot \boldsymbol{u})^2} = \frac{\|\nabla\mathcal{L}(\boldsymbol{x})\|}{d-1}\sin \vartheta. \tag{41}$$

Inserting $|F|$ in Equation (39) yields

$$\nabla \cdot F = -\frac{\|\nabla\mathcal{L}(\boldsymbol{x})\|}{d-1}\frac{(d-1)(\sin \vartheta)^{d-2}\cos \vartheta}{(\sin \vartheta)^{d-2}} = -|\,\|\nabla\mathcal{L}\|\,\boldsymbol{e} \cdot \boldsymbol{u}. \tag{42}$$

## C  Optimal acceptance rate

The optimal acceptance rate argument for MAMS is analogous to the one in Neal (2011). We will use two general properties of the deterministic MH proposal:

1. The expected value of the MH ratio under the stationary distribution, $\mathbb{E}_{z \sim p}[e^{-W(z', z)}]$, is

$$\int p(\boldsymbol{z})e^{-W(\boldsymbol{z}', \boldsymbol{z})}d\boldsymbol{z} = \int p(\boldsymbol{z})\frac{q(\boldsymbol{z}|\boldsymbol{z}')p(\boldsymbol{z}')}{q(\boldsymbol{z}'|\boldsymbol{z})p(\boldsymbol{z})}d\boldsymbol{z} = \int p(\boldsymbol{z}')|\frac{\partial\varphi}{\partial\boldsymbol{z}}(\boldsymbol{z})|d\boldsymbol{z} = \int p(\varphi(\boldsymbol{z}))d\varphi(\boldsymbol{z}) = 1.$$

   This is the Jarzynski equality. In statistical literature it was used by Neal (2011); Creutz (1988) in the special case when $\varphi$ is symplectic.

2. In equilibrium,

$$P(W > 0|\text{accepted}) = P(W < 0|\text{accepted}) = \frac{1}{2}$$

   by the design of the MH algorithm (Neal, 2011). Since $P(\text{accepted}|W < 0) = 1$, we have that

$$\frac{1}{2} = P(w < 0|\text{accepted}) = \frac{P(W < 0|\text{accepted})P(W < 0)}{P(\text{accepted})} = \frac{P(W < 0)}{P(\text{accepted})},$$

   so $P(\text{accepted}) = 2P(W < 0)$.

Let us approximate the stationary distribution over $W$ as $\mathcal{N}(\mu, \sigma^2)$, as in Neal (2011). We then have by the Jarzynski equality:

$$1 = \int p(\boldsymbol{z}) e^{-W(\boldsymbol{z}', \boldsymbol{z})} d\boldsymbol{z} = \int_{-\infty}^{\infty} \frac{1}{\sqrt{2\pi\sigma^2}} e^{-(W-\mu)^2/2\sigma^2} e^{-W} dW = e^{\frac{\sigma^2}{2} - \mu}, \tag{43}$$

implying that $\sigma^2 = 2\mu$. By property (2) we then have

$$P(\text{accept}) = 2\Phi(-\mu/\sqrt{2\mu}), \tag{44}$$

where $\Phi$ is the Gaussian cumulative density function. Denote by $K_{\text{accepted}}$ the number of accepted proposals needed for a new effective sample. This corresponds to moving a distance on the order of the size of the typical set

$$K_{\text{accepted}} N \epsilon \propto \sqrt{d}, \tag{45}$$

since $\sqrt{d}$ is the size of the standard Gaussian's typical set. The number of effective samples per gradient call is then

$$\text{ESS} / \# \text{ gradient evaluations} = \frac{1}{K_{\text{total}} N} = \frac{P(\text{accept})}{K_{\text{accepted}} N} \propto \frac{\epsilon P(\text{accept})}{\sqrt{d}}. \tag{46}$$

The error of the MCHMC Velocity Verlet integrator for an interval of fixed length is (Robnik et al., 2024)

$$\sigma^2/d \propto \epsilon^4/d^2, \tag{47}$$

implying that $\sigma^2 = 2\mu \propto \epsilon^4/d$. Therefore

$$\text{ESS} / \# \text{ gradient evaluations} \propto \mu^{1/4} \, \Phi(-\sqrt{\mu/2}) \, d^{-1/4}, \tag{48}$$

so we see that the efficiency drops as $d^{-1/4}$. ESS is maximal at $\mu = 0.41$, corresponding to $P(\text{accept}) = 65\%$. From Equation (47) we then see that the optimal stepsize grows as $\epsilon \propto d^{1/4}$ instead of the $d^{1/2}$ that would correspond to the unimpaired efficiency.

Note that this result is different if a higher-order integrator is used. For example, when using a fourth order integrator $\sigma^2/d \propto (\epsilon^2/d)^4$, the optimal setting is $\mu = 0.13$ and $P(\text{accept}) = 80\%$.

Empirically, we find that even for a second-order integrator, targeting a higher acceptance rate, of 90%, works well in practice; we use this in our experiments.

## D   Optimal rate of partial refreshments

We will here demonstrate the sensitivity of HMC to trajectory length for ill-conditioned Gaussian distributions and show how Langevin dynamics alleviates this issue. We will then obtain the optimal rate of partial refreshments in the limit of large condition number. For MALT, Riou-Durand and Vogrinc (2022) derive the ESS of the second moments in the continuous-time limit for the Gaussian targets $\mathcal{N}(0, \sigma)$:

$$ESS(\beta, T) = \frac{1 - \rho^2}{1 + \rho^2}, \tag{49}$$

where

$$\rho = e^{-\beta T} \left( \cos \omega T + \frac{\beta}{\omega} \sin \omega T \right) \qquad \omega = \sqrt{\frac{1}{\sigma^2} - \beta^2}. \tag{50}$$

$T$ is the trajectory length, $\beta$ is the LMC damping parameter ($\gamma = 2\beta$ in Riou-Durand and Vogrinc (2022)).

Figure 4 shows ESS as a function of the trajectory length and the rate of partial refreshments, both for the standard Gaussian and for the Gaussian in the limit of very large condition number. For the standard Gaussian, HMC achieves better performance than MALT, but only if trajectory length is chosen well. For ill-conditioned Gaussian however, HMC drastically underperforms compared to MALT. Optimal MALT hyperpatameter settings for the ill conditioned Gaussians are $\beta = 0.567$ and $T = 1.413$. Therefore the optimal ratio of the decoherence time scales of the partial and full refreshment are

$$\frac{t_{\text{partial}}}{t_{\text{full}}} = \frac{1/\beta}{T} = 1.25. \tag{51}$$

We will use the same setting for Langevin MAMS, so $L_{\text{partial}}/L = 1.25$.

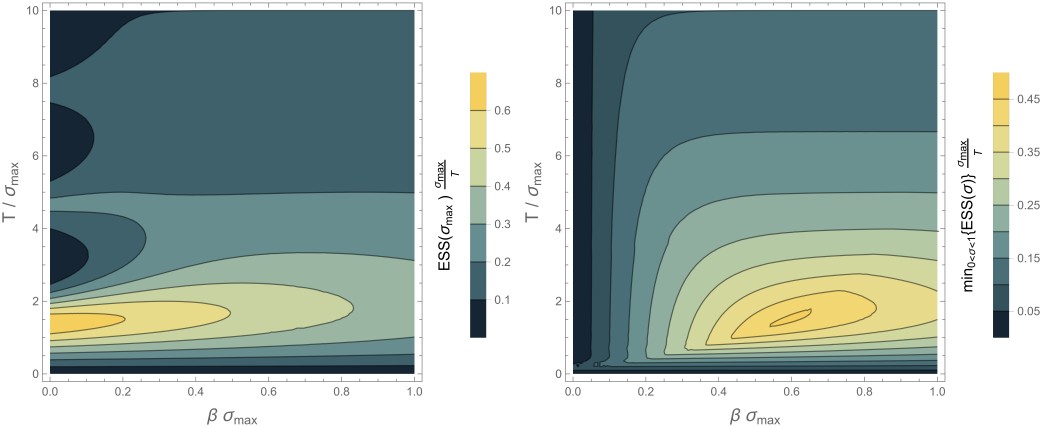

Figure 4: Effective sample size in continuous time for MALT LMC on Gaussian targets. x-axis is the LMC damping parameter, y-axis the trajectory length. $x = 0$ is the HMC line ($\beta = 0$ means there is no damping and no Langevin noise), $x = 1$ is LMC with critical damping $\beta = 1/\sigma_{\max}$. Left panel: isotropic Gaussian $\mathcal{N}(0, \sigma_{\max})$. Note that HMC achieves the optimal performance if properly tuned, the only reason to introduce Langevin noise would be to potentially make the tuning easier. Right panel: extremely ill-conditional Gaussian with all scales $(0, \sigma_{max}]$. ESS along the worst direction is shown. HMC performs poorly as it cannot be tuned to all scales, damping of $\beta\sigma_{max} = 0.57$ performs best. Note that these results do not imply MALT having non-zero ESS in the infinite condition number limit: we only study continuous time MALT here.

# E    Further experimental details

## E.1    Benchmark Inference Models

We detail the inference models used in Section 7. For models adapted from the Inference gym (Sountsov et al., 2020) we give model's inference gym name in the parenthesis.

- Gaussian is 100-dimensional with condition number 100 and eigenvalues uniformly spaced in log.
- Banana (Banana) is a two-dimensional, banana-shaped target.
- Bimodal: A mixture of two Gaussians in 50 dimensions, such that

$$p(\boldsymbol{x}) = (1 - a)\mathcal{N}(\boldsymbol{x}|0, I) + a\mathcal{N}(\boldsymbol{x}|\mu, \sigma^2 I),$$

  where $a = 0.25$, $\mu = (4, 0, \ldots 0)$ and $\sigma = 0.6$.
- Rosenbrock is a banana-shaped target in 36 dimensions. It is 18 copies of the Rosenbrock functions with Q = 0.1, see ().
- Cauchy is a product of 100 1D standard Cauchy distributions.
- Brownian Motion (BrownianMotionUnknownScalesMissingMiddleObserva-tions) is a 32-dimensional hierarchical problem, where Brownian motion with unknown innovation noise is fitted to the noisy and partially missing data.
- Sparse logistic regression (GermanCreditNumericSparseLogisticRegres-sion) is a 51-dimensional Bayesian hierarchical model, where logistic regression is used to model the approval of the credit based on the information about the applicant.
- Item Response theory (SyntheticItemResponseTheory) is a 501-dimensional hier-archical problem where students' ability is inferred, given the test results.
- Stochastic Volatility is a 2429-dimensional hierarchical non-Gaussian random walk fit to the S&P500 returns data, adapted from numpyro (Phan et al., 2019b).
- Neal's funnel (Neal, 2011) is a funnel shaped target with a hierarchical parameter $z_1 \sim \mathcal{N}(0, 3)$ that controls the variance of the other parameters $z_i \sim \mathcal{N}(0, e^{z_1/2})$ for $i = 2, 3, \ldots d$. We take $d = 20$.

|               | MAMS   | MAMS ($\times$ 10) | MAMS (/ 10) |
|---------------|--------|--------------------|-------------|
| Gaussian      | 3249   | 3162               | 3169        |
| Banana        | 14,078 | 13,098             | 13,469      |
| Rosenbrock    | 94,184 | 91,620             | 94,533      |
| Brownian      | 13,528 | 13,759             | 13,973      |
| German Credit | 55,748 | 59,777             | 57,118      |
| ItemResp      | 45,371 | 43,840             | 450,40      |

Table 2: Number of gradients calls needed to get the squared error on the worst second moment below 0.01. MAMS is compared with MAMS where the initial step size has been multiplied (third column) or divided (fourth column) by a factor of 10. This demonstrates that the performance of the automatic hyperparameter adaptation is not sensitive to the initialization.

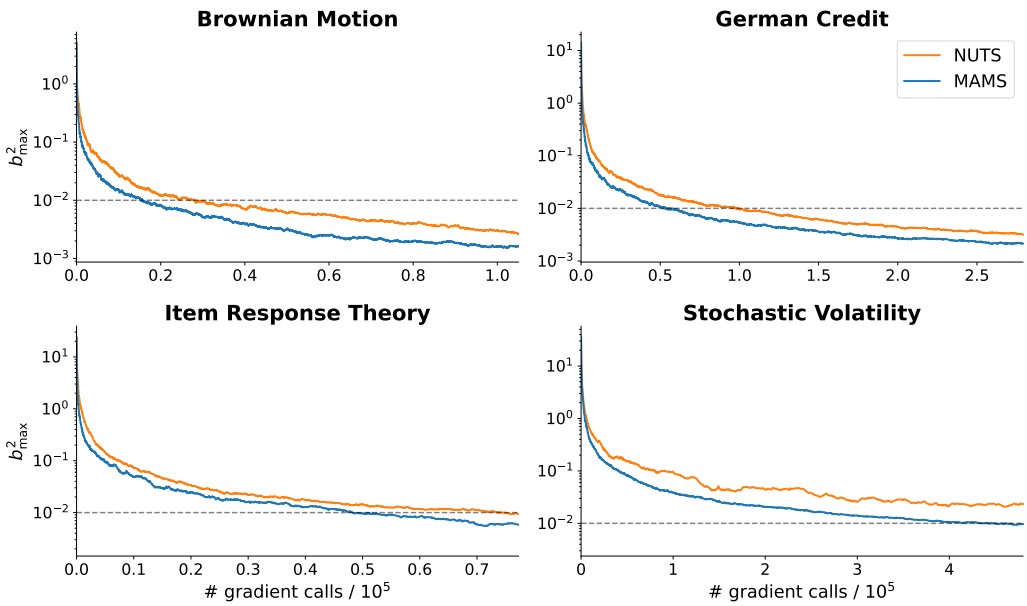

Figure 5: Sampling error, as measured by $b_{\max}^2$, as a function of computational cost, as measured by number of gradient calls. MAMS is compared with NUTS on several benchmark problems, demonstrating it consistently outperforms NUTS. Gradient calls to $b_{\max}^2 = 0.01$ dashed line is the number reported in Table 1.

Ground truth expectation values $\mathbb{E}[x^2]$ and $\mathrm{Var}[x^2] = \mathbb{E}[(x^2 - \mathbb{E}[x^2])^2]$ are computed analytically for the Ill Conditioned Gaussian, by generating exact samples for Banana, Rosenbrock and Neal's funnel and by very long NUTS runs for the other targets.

## E.2 Sensitivity to the hyperparameter initialization

MAMS hyperparameters are automatically determined by the algorithm from Section 6. However, the algorithm is initialized with some stepsize $\epsilon_{init} = 0.2\sqrt{d}$ and trajectory length $L_{init} = \sqrt{d}$. MAMS performance could in principle be sensitive to this initialization. In Table 2 we show that this is not the case.

## E.3 Convergence curves

Figure 5 shows the convergence of the second moments as a function of gradient calls used for NUTS and MAMS. $b_{\max}^2$ is computed as in Section 7. Sampling error for both samplers steadily decreases, but MAMS is consistently faster.

| Model | Metric | MAMS | NUTS |
|---|---|---|---|
| Banana | $b_{max}$ | 0.42% | 1.52% |
| Bimodal Gaussian | $b_{max}$ | 6.34% | 2.07% |
| Brownian Motion | $b_{max}$ | 0.36% | 0.22% |
| Cauchy | $b_{avg}$ | 5.37% | 0.02% |
| German Credit | $b_{max}$ | 0.12% | 0.09% |
| Item Response | $b_{max}$ | 1.05% | 0.17% |
| Rosenbrock | $b_{avg}$ | 0.06% | 0.02% |
| Standard Gaussian | $b_{avg}$ | 1.04% | 0.13% |
| Stochastic Volatility | $b_{max}$ | 0.10% | 0.09% |

Table 3: Relative uncertainty associated with Table 1.

### E.4 Uncertainty in Table 1

Table 3 shows the relative uncertainty in the results from Table 1. Uncertainty is calculated by bootstrap: for a given model, we produce a set of chains (usually 128), and calculate the bias $b_{max}$ at each step of the chain. We then resample (with replacement) 100 times from this set, and compute our final metric (number of gradients to low bias) 100 times. We take the standard deviation of this list of length 100 to obtain an estimate of the error. This error, relative to the values in Table 1 is then reported in percent.

### E.5 Computational architecture

The experiments were run on 128 CPU cores, where each core is a 2x AMD EPYC 7763 (Milan) CPU.

