# OpenReview forum: "Metropolis Adjusted Microcanonical Hamiltonian Monte Carlo"
_NeurIPS.cc/2025/Conference — NeurIPS 2025 poster_

### Official Review · Reviewer_rovM · 2025-06-28

**Clarity:** 1
**Significance:** 3
**Originality:** 3
**Rating:** 4
**Confidence:** 3

**Summary:**

The paper proposes MAMS, a Metropolis-adjusted micro-canonical sampler that improves sampling efficiency in high dimensions. It corrects the bias in MCLMC while maintaining its efficiency and requires no manual tuning. MAMS also outperforms NUTS by achieving speedup empirically.

**Questions:**

- In eq (1), is the Markov chain $x\rightarrow x'$ or $x'\rightarrow x$?
- The authors should define $d$ to be the dimension.
- In eq (4), why do you need to flip the sign of the velocity after $n$-th leapfrog integration?
- What is the explicit form of potential energy $V$ and kinetic energy $K$ in line 121? Are $V(x)=\mathcal{L}(x)$ and $K(u)=\frac{1}{2}||u||^2$? If so, these should be clearly defined.
- In eq (9), does $\delta(\varphi(z)-z')$ mean $\delta$ multiplied by $\varphi(z)-z'$ or $\delta$ evaluated at $\varphi(z)-z'$? If the latter is true, $\delta$ in line 125 should be defined as a function of $x$, i.e., $\delta(x)$.
- In line 122, why do you also update the velocity to $u'$ by the position map $A_{\epsilon}$?
- Why is $\varphi$ defined differently in Lemma 4.2 compared to Equation (4)?
- What is the explicit form of $\delta(z',z)$ in line 207?
- What is ESS in line 279? Is it defined or referred in this paper?
- It would be more insightful if the authors provided convergence curves for all samplers involved in the experiment to enable a better comparison.

**Ethical Concerns:**

["NO or VERY MINOR ethics concerns only"]

**Final Justification:**

The authors convince me that the derivation of a valid Metropolis-Hastings rule for microcanonical dynamics is nontrivial. The experiment shows that it consistently outperforms NUTS in multiple settings. Thus I would like to raise my score to 4.

**Limitations:**

No. The authors claim that the limitation is mentioned in experiments and discussed in the conclusions, but it remains unclear what the specific limitation is. It would be helpful to present it more explicitly, preferably in a dedicated paragraph or section.

**Quality:**

2

**Strengths And Weaknesses:**

**Strength:**
- This paper studies a novel sampler-MAMS, which incorporates Metropolis-Hasting filter to MCLMC to correct bias and exhibits empirical acceleration over NUTS. It also introduces adaptive step size and trajectory length to enhance practicality and ease of use.

- Extensive numerical experiments demonstrate the advantages of MAMS over NUTS.

**Weakness:**
- This paper is not well-written with several confusing and unclear notations (see Questions below). One of its main contributions—the proposed algorithm—is deferred to the appendix, which significantly hinders readability and understanding.
- The theoretical contribution of this paper is marginal: while [1] provides a mixing analysis of NUTS for Gaussian target distributions, this paper lacks analysis of the mixing properties of MAMS.

[1]Bou-Rabee, N., & Oberdörster, S. (2024). Mixing of the No-U-Turn Sampler and the Geometry of Gaussian Concentration. arXiv preprint arXiv:2410.06978.

---

> ### Author Rebuttal · Authors · 2025-07-31
>
> We thank the reviewer for their time and thoughtful feedback. We understand that the main concerns relate to the lack of theoretical mixing guarantees and aspects of clarity in notation and presentation. We address both below.
>
> ### Theoretical Contribution and Significance
> We respectfully disagree with the assessment that this work lacks substantial theoretical contribution. A key theoretical novelty of our paper is the **derivation of a valid Metropolis-Hastings acceptance rule for microcanonical dynamics**, enabling the first **asymptotically unbiased sampler** based on MCLMC. This derivation is nontrivial, especially for MAMS with Langevin noise, as a standard procedure needs to be generalized to non-volume preserving dynamics.
> While we do not include formal mixing-time bounds (as in [1]), our primary goal is **practical performance**. As NeurIPS emphasizes empirical relevance and impact across ML subfields, we focus on showing that **MAMS consistently outperforms NUTS** — the de facto standard black-box sampler — across benchmark problems. We also study its scaling with the dimensionality and condition number. This addresses a real need in practical sampling and represents an important advance for the applied ML and probabilistic programming communities.
> Mixing analysis, especially in the microcanonical setting, is an interesting and challenging direction, and we agree that this is valuable future work. However, we believe that the current results — both theoretical and empirical — already justify publication. We ask the reviewer to consider this framing and the practical significance of our contribution in the NeurIPS context.
>
> ### Presentation and notation
> We appreciate the detailed list of clarity issues and have addressed them point-by-point below. We also note that other reviewers found the presentation strong (e.g., Reviewer DLZV: *"All the results are stated very clearly, without overloaded notation, and with clear contributions."*). Nevertheless, we fully agree that papers must be accessible across subfields and have revised the manuscript accordingly:
> - x -> x’, we have clarified this in the text.
> - Although not explicitly stated, d can be seen to be the dimension in the first line of the introduction, where we state, $x \in \mathbb{R}^d$. We have explicitly stated that it is the dimension.
> - Velocity flip is necessary to make the proposal map $\varphi$ an involution. Indeed its effect is erased by the full refreshment, but we still keep it for consistency.
> - MCLMC dynamics are a time-rescaled Hamiltonian dynamics, so the kinetic energy difference between two states z1 and z2 cannot be defined as K(z2) - K(z1), but is instead a nonlinear function K(z2, z1). This is why we have not directly defined K(z), but only its difference between the states. Note that it is only the difference that is relevant for the acceptance probability. We have now clarified this around Equations 6, 7 and 8.
> - $\delta$ is a delta function so yes, it can be read as being evaluated at its argument (with all the usual subtleties of the delta “function”). We realize that the proof of Lemma 4.1 with delta functions might not appeal to the mathematically inclined audience, so we have now also included the reference to the other proofs from the literature, for example [1].
> - Thank you for catching this, it was a typo.
> - $\varphi$ in Equation 4 is defined by a leapfrog update, in Lemma 4.2 it is slightly more general, allowing for other split-type integrators. Thank you for pointing this out, we have clarified it in the text.
> - Indeed this is a very sloppy notation, because $\Delta$ does not actually depend on z and z’, but rather only on z’’ = O(z). We instead should have defined $\Delta$(z’’) = K(BAB(z’’)) - K(AB(z’’)) + V(AB(z’’)) - V(B(z’’)) + K(B(z’’)) - K(z’’), we have now fixed this in the text.
> - Thank you for catching this, we have indeed not defined it. We define Effective sample size (ESS) as: number of samples / $\tau_{int}$, from Equation 13. This is very standard, see for example [2].
> - We agree this would be a nice illustration, we have added these curves to the appendix.
>
> [1] Neklyudov, Kirill, et al. "Involutive MCMC: a unifying framework." *International Conference on Machine Learning*. PMLR, 2020.
> [2] Stan reference manual, Section 15.4: Effective sample size.
>
> ### Limitations section
> From a practitioner’s standpoint, a limitation of MAMS compared to NUTS is that it has not been so widely tested. This is understandable, given that we have just proposed MAMS, while NUTS has been state-of-the-art for years. This limitation is highlighted in the discussion.

---

> > ### Comment · Reviewer_rovM · 2025-08-04
> >
> > I thank the authors for their thoughtful rebuttal, which addresses most of my questions. I believe that the derivation of a valid Metropolis-Hastings rule for microcanonical dynamics is nontrivial and enables the first unbiased sampler in the MCLMC family. While some theoretical aspects remain open, I am happy to update my score.

---

### Official Review · Reviewer_2xBS · 2025-07-01

**Clarity:** 3
**Significance:** 3
**Originality:** 3
**Rating:** 5
**Confidence:** 5

**Summary:**

This paper designs a new black-box sampler leveraging recent developments with a new type of "microcanonical" dynamics for efficient exploration.
Bias is controlled by deriving the Metropolis Hastings acceptance rate for these dynamics.
An automatic hyper-parameter selection is designed, the analogue of NUTS for HMC, so that practitioners can use the sampler as a black-box without tuning.
The authors make the code available with a full featured sampler comparison suite.

**Questions:**

## Micro-canonical versus canonical

> We refer to our sampler as using microcanonical dynamics in reference to previous work on MCLMC, In contrast, we will refer to the dynamics in standard HMC as canonical dynamics.

It's not critical but maybe you could give a little intuition to readers about these names. (E.g. most samplers are motivated by designing physical dynamics which match the target in thermal equilibrium, called the canonical ensemble in physics, while the proposed samplers were motivated by designing an energy preserving dynamics, or microcanonical ensemble in physics, that directly matches the target.)
The intro for Eq. 2 and 3 makes them look so similar it's not obvious where the difference comes from. (I like that the notation shows how effectively similar HMC and microcanonical are.)

- The final presentation of the MH weight was not very readable. I guess one proceeds from Lemma 4.2, replacing expressions with Eq. 6 & 8. It goes through some intermediate notation about the steps. I get that the substep notation is important, to know which energy differences are incorporated, but it makes it hard to read. If it's possible, it would be nice to have a summary in the main text which summarizes the step and final weight, without intermediate notation.

- Using NUTS for preconditioning is kind of disappointing, if the goal is to replace NUTS

- Trajectory length. I didn't follow this part very well, though the intuition makes sense (to make trajectories long enough to reduce autocorrelation). Was this idea explored before, or is it similar logic to NUTS? My memory of NUTS is dim - obviously "No U-Turn" is the criteria, but I thought they determined that in a more dynamics-focused way, not based on autocorrelation. It just wasn't clear to me where your approach came from. (Obviously it works well, though!)


- Footnote 2 "Note that the dynamics... are not Hamiltonian" was confusing, I think you just wanted to point out that the kinetic energy for MCHMC is not standard. It's still "Hamiltonian dynamics", with a non-standard kinetic energy

- You mentioned involution a few times. Like other potential readers, I was not familiar with the term before I had read this nice paper on the useful properties of involutions in sampling.
http://proceedings.mlr.press/v119/neklyudov20a/neklyudov20a.pdf

**Ethical Concerns:**

["NO or VERY MINOR ethics concerns only"]

**Final Justification:**

I maintained my original score to accept.
- Authors did a good job with follow-ups. The extensive supplementary actually addressed some issues, and they have promised to clarify in text where possible.
- I didn't see any convincing arguments in the low score reviews. One reviewer found the paper hard to follow. I didn't find it hard to follow, but that may be because I was more familiar with papers in this specific area.

**Limitations:**

yes

**Quality:**

4

**Strengths And Weaknesses:**

Strengths:

- NUTS has been very influential because of its solid, hands-free sampling performance. Getting significant improvements with new dynamics while maintaining the automated, black-box usage is a significant advance.
- Nice derivation of the MH correction for the microcanonical sampling dynamics, which to my knowledge has not appeared before in this literature.
- I appreciate that the authors did not take for granted some of the developments that applied to HMC (like whether to do partial or full moment refresh, what acceptance rate is ideal), and investigated these aspects.
- It seems like best practices from the literature were incorporated at every level, including little tricks I wasn't familiar with, like the step number randomization (Neal's original HMC paper mentioned the potential issue, but gave no solution).
- The automatic hyper-parameter selection scheme, especially the trajectory length part, does not look trivial to come up with. The results comparing to grid search show that it is very effective.
- It looks like it's being released with a pretty full-featured JAX sampling library, which is a great resource for the field.
- Solid, well-thought out experiments


Weaknesses:

The paper is technically heavy and presumes some deep background about samplers. I don't see any way around this, but I use the questions section to suggest areas where clarity may be improved.

---

> ### Author Rebuttal · Authors · 2025-07-31
>
> We sincerely thank the reviewer for the thoughtful and encouraging feedback. We address the points below.
> ### Microcanonical vs. Canonical — Physical Background
> We agree that it would be helpful to provide a clearer explanation of the physical intuition behind the equations 2 and 3 in the main text. While we discuss this in Appendix B, we now expand on it in the main text, as suggested.
> ### MH weight
> Thank you for pointing this out, we understand a clear statement of the final MH weight is valuable, especially if one wants to implement the algorithm. In simple terms, for every B or A update that the algorithm performs in the proposal, the energy difference must be calculated and added to the cumulative energy difference (B update energy difference is calculated by Equation 8, A update by Equation 6). Final MH acceptance probability is min(1, e^-total cumulative energy difference). We have clarified this in the paper and also moved the algorithm’s pseudocode to the main text.
> ### Use of NUTS for Preconditioning
> We agree that relying on NUTS for preconditioning may seem contradictory to our goal. In our experiments, we opted for NUTS-based preconditioning solely to isolate the sampler's ability to sample from its ability to precondition (which, granted, are strongly related). Preconditioning can be viewed as reparametrizing the parameter space, so we wanted to compare NUTS with MAMS on exactly the same problem, i.e., same target, same parametrization.
> In practice, however, we recommend using MAMS for preconditioning as well which is in fact the default behavior in our released implementation. We have clarified this in the updated manuscript.
> ### Trajectory Length Adaptation
> The trajectory length adaptation used in MAMS is not related to NUTS, but instead follows the approach proposed in [1] for unadjusted MCLMC. We have adapted it here in a natural way to work with the Metropolized variant. We now state this more clearly in the text.
> ### Equation 3 and Hamiltonian Dynamics
> Technically, Equation 3 does not describe Hamiltonian dynamics in the usual sense, as the transformation is not volume-preserving (the Jacobian is not unity). However, it does represent reparametrized Hamiltonian dynamics, where the time has been rescaled such that the velocity has unit norm. This connection allows us to define the change in energy along these trajectories, as discussed in Appendix B. We have revised the footnote to make this distinction more precise.
> ### Involutive MCMC Reference
> We appreciate the pointer to Neklyudov et al. (2020). We now cite this reference in the relevant section and briefly discuss its connection. Specifically, MAMS without Langevin noise is compatible with the involutive MCMC framework, while the stochasticity introduced by Langevin noise moves it outside of the class of deterministic involutive proposals.
>
> Once again, we thank the reviewer for the detailed and constructive suggestions. We believe these clarifications will improve the paper’s readability and accessibility.
>
> [1] Robnik, J., G. B. De Luca, E. Silverstein, and U. Seljak (2024). Microcanonical Hamiltonian Monte Carlo. The Journal of Machine Learning Research 24(1), 311:14696–311:14729

---

### Official Review · Reviewer_K5dp · 2025-07-02

**Clarity:** 3
**Significance:** 3
**Originality:** 3
**Rating:** 4
**Confidence:** 2

**Summary:**

The paper introduces a new version of mcmc method designed for efficient sampling from high-dimensional probability distributions. It is built on MCLMC by adding a MH correction step, removing the asymptotic bias. In addition , the paper proposes a fully automated hyperparameter tuning scheme for general use. The method is implemented in JAX and will be open-sourced.

**Questions:**

1. Can you provide results or qualitative comparisons to other gradient-based samplers?
2. I would like the authors to highlight the novelty of the paper from either empirical or theoretical perspective.
3. I wonder how sensitive is MAMS to poor step size or trajectory length initialization during tuning?

**Ethical Concerns:**

["NO or VERY MINOR ethics concerns only"]

**Final Justification:**

My primary questions and follow-up questions are addressed by the reviewers. The authors admit that the method requires much more evidence to be successor to NUTs. Given the overall novelty and theoretical contribution, I’m willing to raise my score.

**Limitations:**

no. I don't think this work has potential negative societal impact.

**Quality:**

3

**Strengths And Weaknesses:**

This paper presents MAMS, which applies a MH correction to MLMC to produce unbiased samples with efficiency advantages. Rigorous derivation of the acceptance rule is provided and implement a practical automatic tuning scheme. Empirical results show significant improvements over NUTS across a range of benchmarks.

My main concern is the novelty of the method. The core contribution seems to be an integration of existing ideas. In addition, the experimental comparison is restricted to NUTS.

---

> ### Author Rebuttal · Authors · 2025-07-31
>
> We thank the reviewer for their thoughtful comments and for recognizing the technical soundness, clarity, and empirical strength of our work. We’d like to respond to the concern regarding novelty and the scope of comparisons.
>
> ### Novelty and Significance
> Our main contribution is the development of a new black-box gradient-based sampler that is **asymptotically unbiased** and consistently outperforms NUTS — currently the most widely used black-box MCMC method (e.g., in Stan [1]). We believe this alone is a very significant result, with a direct impact for practitioners. From a theoretical standpoint, the derivation of the MH acceptance rule for our microcanonical sampler (with and without Langevin noise) is non-trivial and novel. These derivations are necessary to ensure unbiasedness. Additionally, we propose a new automatic hyperparameter tuning scheme — including for trajectory length — which is key to making the sampler practical. One reviewer remarked that "best practices from the literature were incorporated at every level, including little tricks I wasn't familiar with..." and highlighted the tuning scheme as a non-obvious, effective contribution. We believe this underscores the care and originality in the algorithm’s design.
>
> ### Comparison to Other Methods
> Regarding the scope of baselines, we emphasize that our goal is to present a **general-purpose black-box sampling scheme**. While many gradient-based samplers exist in the literature, **very few include both a sampler and a tuning mechanism** that can be used out of the box — NUTS is the main such baseline. We therefore chose to focus on this meaningful and widely adopted comparator.
> Adding comparisons to methods that require significant hand-tuning would not be fair or informative in this context. That said, if the reviewer has specific methods in mind that meet the same black-box usability standard, we would be happy to discuss them.
>
> ### Sensitivity to Initialization
> We appreciate the question regarding sensitivity to step size and trajectory length initialization. We have now added a robustness experiment to the appendix: we repeated the tuning procedure using initial step sizes and $L$ values that were 10× larger and 10× smaller. The resulting tuned values and performance metrics (Table 1) were nearly unchanged, demonstrating that our scheme is **not sensitive to initialization**.
>
> ### Conclusion
> We hope this clarifies the novelty and significance of our contributions. In summary, the proposed method offers a new black-box unbiased algorithm that outperforms NUTS. Developing it required non-trivial derivation of the Metropolis-Hastings acceptance probability and a combination of tricks to make it black-box. The resulting algorithm is efficient, robust and applicable out of the box, and is intended to serve as a successor to NUTS. We believe these advances justify the paper's relevance and significance, and we respectfully ask the reviewer to consider revising their score in light of this clarification.
>
> [1] Stan: A Probabilistic Programming Language, Bob Carpenter, Andrew Gelman, Matthew D. Hoffman, Daniel Lee, Ben Goodrich, Michael Betancourt, Marcus Brubaker, Jiqiang Guo, Peter Li, Allen Riddell

---

> > ### Comment · Reviewer_K5dp · 2025-08-04
> > **Follow-up questions**
> >
> > To me, MH correction is a standard recipe in the sense that for any reversible proposal, one can usually compute the acceptance probability from the target density and the Jacobian of the transformation. Simply asserting that it is non-trivial or quoting a reviewer’s praise is not a valid response. Another concern is that automatic hyperparameter tuning that makes MAMS “black-box” like NUTS is not convincing enough to show the method can serve as a successor to NUTS, which is why I’m asking for more robustness experiments/empirical evidence of effectiveness on real use cases. NeurIPS rebuttal this year still supports authors sharing result in table format and I would appreciate authors to share results directly instead of promising to add them into appendix.

---

> > > ### Author Response · Authors · 2025-08-04
> > >
> > > ### MH ratio for deterministic dynamics
> > >
> > > In the case of microcanonical dynamics as defined in Robnik et al 2022, Metropolis-Adjustment is ill defined, because the target distribution is a delta function at some energy level, therefore any proposal that does not conserve the energy would be rejected. We therefore have to make use of a rescaling (proposed in many places, including [2] but reviewed in detail in appendices B1 and B2), which obtains the dynamics given in equation (3) of the paper.
> > >
> > > As you say, calculating the MH ratio now amounts to calculating the Jacobian, but in this case, the Jacobian is not particularly easy to compute. In lemma 4.2, we show how this can be done. As the reviewer may note, this makes use of a series of equalities, several of which are derived in other papers, but we don't feel that the derivation as a whole is obvious.
> > >
> > > Moreover, our derivation reveals a surprising fact. As we show in appendix B5 that, when viewing the dynamics of equation (3) as a rescaling of a Hamiltonian system with log-kinetic energy, W, defined by a=min(1, e^{-W}), is nothing but the energy change of the original Hamiltonian.
> > >
> > > ### Extension to Langevin dynamics: Theorems 5.1 and 5.2
> > >
> > > We'd also like to highlight our proof that the same acceptance probability for the kernel holds when Langevin noise is present on the momentum. This is theorem 5.1, which is proven in appendix A1. Further, we show that the analogue of the Metropolis Adjusted Langevin Trajectories (MALT) approach of [3] works in this setting (theorem 5.2, proven in appendix A2).
> > >
> > > We think both of these proofs (appendices A1 and A2) represent theoretical contributions.
> > >
> > > Finally,
> > > Simply asserting that it is non-trivial or quoting a reviewer’s praise is not a valid response.
> > >
> > > That's fair enough. We hope the above is more informative - please let us know if clarification is needed.
> > >
> > > [2] Hamiltonian Dynamics with Non-Newtonian Momentum for Rapid Sampling. Greg Ver Steeg, Aram Galstyan
> > > [3] Metropolis Adjusted Langevin Trajectories: a robust alternative to Hamiltonian Monte Carlo. Lionel Riou-Durand and Jure Vogrinc
> > >
> > > ### Real-world examples
> > > > Another concern is that automatic hyperparameter tuning that makes MAMS “black-box” like NUTS is not convincing enough to show the method can serve as a successor to NUTS, which is why I’m asking for more robustness experiments/empirical evidence of effectiveness on real use cases. NeurIPS rebuttal this year still supports authors sharing result in table format and I would appreciate authors to share results directly instead of promising to add them into appendix.
> > >
> > > We're not clear here on whether the reviewer views the cases that we've studied as real use cases or not. For example, table 1 of the paper shows the results on a range of distributions, including the Stochastic Volatility model, which is a model of real data. Is this the kind of thing the reviewer has in mind?
> > >
> > > Certainly we agree that much more evidence is needed on real-world applications, but this is a many-year undertaking that requires a community of users (such as the Stan userbase). We see this paper as a first step in getting there, and think the evidence we provide is already quite strong. We’d like to draw the reviewer’s attention to our comparison of our automatic tuning with grid search results (last column in table 1), which suggests that the tuning is robust.

---

> > > > ### Comment · Reviewer_K5dp · 2025-08-04
> > > >
> > > > Thank you for the response. My questions are clear and I will adjust my scores accordingly.

---

### Official Review · Reviewer_DLZV · 2025-07-03

**Clarity:** 4
**Significance:** 3
**Originality:** 3
**Rating:** 5
**Confidence:** 4

**Summary:**

Microcanonical Langevin Monte Carlo (MCLMC) offers an efficient alternative to Hamiltonian Monte Carlo (HMC) on multiple target distributions. However, like for HMC, the produced samples converge to the correct distribution only for the continuous-time limit. To address this, HMC employs the Metropolis-Hastings test, which removes the bias by guaranteeing that the corresponding Markov chain preserves the target distribution. MCLMC has been missing such a test, and the current paper closes this gap. In particular, it designs the Metropolish-Hastings-Green test [1], taking the Microcanonical Langevin dynamics integrated in time as a proposal. Furthermore, the authors propose practical ways for choosing and tuning hyperparameters. Finally, they empirically study the algorithm's performance for various target distributions.

[1] Green, Peter J. "Reversible jump Markov chain Monte Carlo computation and Bayesian model determination." Biometrika 82, no. 4 (1995): 711-732.

**Questions:**

I have no questions or suggestions that would affect my evaluation. Personally, I'm wondering if the result of Theorem 5.1 is similar to the framework proposed in [4]. Not the evaluation of the MCLMC Jacobian, but the test of a non-measure-preserving involutive function.

[4] Neklyudov, Kirill, Max Welling, Evgenii Egorov, and Dmitry Vetrov. "Involutive MCMC: a unifying framework." In International Conference on Machine Learning, pp. 7273-7282. PMLR, 2020.

**Ethical Concerns:**

["NO or VERY MINOR ethics concerns only"]

**Final Justification:**

The paper has a solid theoretical contribution and a good empirical study meeting all the criteria for a NeurIPS publication. The rebuttal hasn't changed my evaluation.

**Limitations:**

The authors study and discuss limitations of their work throughout the paper.

**Paper Formatting Concerns:**

The paper formatting does not raise any concerns.

**Quality:**

4

**Strengths And Weaknesses:**

The paper presents a complete, novel, and relevant study. Especially, I would like to highlight the following points.
1. All the results are stated very clearly, without overloaded notation, and with clear contributions.
2. The motivation behind the proposed test is very clear and is additionally highlighted in Fig. 1.
3. Besides their theoretical contribution, the authors provide practical guidelines, whose importance is hard to overestimate.
4. The empirical study is well-motivated, and the choice of metrics, baselines, and benchmarks is very well explained.
In general, the paper meets the standards of a NeurIPS publication.

In my opinion, the paper does not have major weaknesses. However, I would suggest adding the applications and benchmarks that are more interesting for the NeurIPS audience. Indeed, in recent years, the community has transitioned from classical Monte Carlo benchmarks [2] (as considered in the current paper) to applied problems in the natural sciences [3]. I believe that this would significantly strengthen the paper.

[2] Vargas, Francisco, Will Grathwohl, and Arnaud Doucet. "Denoising diffusion samplers." arXiv preprint arXiv:2302.13834 (2023).
[3] Akhound-Sadegh, Tara, Jarrid Rector-Brooks, Avishek Joey Bose, Sarthak Mittal, Pablo Lemos, Cheng-Hao Liu, Marcin Sendera et al. "Iterated denoising energy matching for sampling from Boltzmann densities." arXiv preprint arXiv:2402.06121 (2024).

---

> ### Author Rebuttal · Authors · 2025-07-31
>
> We thank the reviewer for their thoughtful and encouraging comments. We especially appreciate the recognition of the clarity, practical relevance, and completeness of our contributions.
>
> We agree that including more applied or domain-specific benchmarks, particularly those relevant to the natural sciences would further strengthen the impact of our work. We see this as a valuable direction for future research and plan to explore it in subsequent work.
>
> Regarding the connection to [4], indeed, **MAMS without Langevin noise** fits within the involutive MCMC framework presented in [4], as it involves a deterministic, involutive map. However, **MAMS with Langevin noise** falls outside this framework, as the Langevin proposal introduces stochasticity, which violates the determinism requirement in [4]'s construction. We have added an appropriate reference to [4].

---

> > ### Comment · Reviewer_DLZV · 2025-08-04
> >
> > I acknowledge the rebuttal, and I'm keeping my score.
> >
> > Regarding the presence of noise in [4]. I believe that [4] incorporates the noise by extending the state space. In particular, it describes Hamiltonian Monte Carlo and adds the direction variable similarly to the proposed algorithm.

---

> > > ### Author Response · Authors · 2025-08-05
> > >
> > > It does include the full refreshment noise at the end of the proposal trajectory (as in HMC), but not the partial refreshment noise during the proposal trajectory (as in LMC or MCLMC).

---

### Decision · Program_Chairs · 2025-09-17

**Decision:**

Accept (poster)

**Comment:**

The paper extends microcanonical Langevin Monte Carlo (MCLMC) that was proposed as an alternative to Hamiltonian Monte Carlo (HMC). The main and significant contribution is to add a Metropolis-Hastings correction step in order to ensure that the
sampler converges to the target distribution. Adding such correction step requires to compute the ratio of the densities, which was not easy,  and the authors have shown how to do this.

All reviewers agreed that this is a good contribution to the MCMC literature. The paper is well-written and the experiments show
the method can work better than NUTS.

I do feel though that in the experimental comparison a couple of obvious (fully black box gradient-based samplers) should have been included and I will encourage the authors to add them for the final versions (reviewer K5dp also asked about this).

The first obvious addition is standard preconditioned MALA:
$$
q(x'|x) = N(x' | x - \frac{\sigma^2}{2} S \nabla L(x), \sigma^2 S)
$$
where $S$ should be the same preconditioner used by NUTS and MAMS. Given $S$, $\sigma^2$ is further tuned
to achieve acceptance rate $57.4$\%. Obviously this is completely black box.  This MALA is like HMC with L=1, and
you could also add MAMS-MALA by just setting L=1 in MAMS and again tune the step size targeting acceptance rate  around 57.4.
Both samplers require only one gradient evaluation per MCMC iteration.